UPDATE ARTICLE

# Integrin–ECM interactions and membrane-associated Catalase cooperate to promote resilience of the *Drosophila* intestinal epithelium

**Mohamed Mlih**◉*, **Jason Karpac**◉*

Department of Molecular and Cellular Medicine, Texas A&M University, College of Medicine, Bryan, Texas, United States of America

* mmlih@tamu.edu (MM); karpac@tamu.edu (JK)

The Editors encourage authors to publish research updates to this article type. Please follow the link in the citation below to view any related articles.

## Abstract

Balancing cellular demise and survival constitutes a key feature of resilience mechanisms that underlie the control of epithelial tissue damage. These resilience mechanisms often limit the burden of adaptive cellular stress responses to internal or external threats. We recently identified Diedel, a secreted protein/cytokine, as a potent antagonist of apoptosis-induced regulated cell death in the *Drosophila* intestinal midgut epithelium during aging. Here, we show that Diedel is a ligand for RGD-binding Integrins and is thus required for maintaining midgut epithelial cell attachment to the extracellular matrix (ECM)-derived basement membrane. Exploiting this function of Diedel, we uncovered a resilience mechanism of epithelial tissues, mediated by Integrin–ECM interactions, which shapes cell death spreading through the regulation of cell detachment and thus cell survival. Moreover, we found that resilient epithelial cells, enriched for Diedel–Integrin–ECM interactions, are characterized by membrane association of Catalase, thus preserving extracellular reactive oxygen species (ROS) balance to maintain epithelial integrity. Intracellular Catalase can relocalize to the extracellular membrane to limit cell death spreading and repair Integrin–ECM interactions induced by the amplification of extracellular ROS, which is a critical adaptive stress response. Membrane-associated Catalase, synergized with Integrin–ECM interactions, likely constitutes a resilience mechanism that helps balance cellular demise and survival within epithelial tissues.

## Introduction

The evolution of organ systems and specified tissues in metazoans has been accompanied by unique mechanisms that enable these tissues to limit damage caused by external or internal threats. Most tissues thus employ mechanisms to both resist and tolerate such threats. While adaptive cellular stress responses are initiated to antagonize (or resist) these threats, tolerance (or resilience) is a strategy to reduce the negative impact of the same responses. Many cellular stress responses (such as inflated temperature, oxidative stress, or elevated protein folding

**Data Availability Statement:** All relevant data are within the paper and its Supporting Information files.

**Funding:** This work was in part supported by an American Federation for Aging Research Grant (to J.K., https://www.afar.org). The funders had no role in study design, data collection and analysis, decision to publish, or preparation of the manuscript.

**Competing interests:** The authors have declared that no competing interests exist.

**Abbreviations:** EB, enteroblast; EC, enterocyte; ECM, extracellular matrix; EE, enteroendocrine; ERK, extracellular signal–related kinase; ISC, intestinal stem cell; JNK, Jun-N-terminal kinase; MARCM, mosaic analysis with a repressible cell marker; MEK, mitogen-activated ERK kinase; RCD, regulated cell death; ROS, reactive oxygen species.

stress) therefore operate at the cost of normal tissue function and can elicit damage. Resilience mechanisms, which limit the burden of these adaptive stress responses, can in part enhance tolerance to tissue damage through protection and/or repair [1]. These same resilience mechanisms can also be hijacked by cancer cells to promote their survival [2] and by viruses to enhance replication/abundance [3].

Unique tissues have unique tolerance capacities. Epithelial tissues, which act as a barrier to the external environment and are often capable of renewal, generally have a high tolerance capacity and employ wide-ranging resilience mechanisms to limit damage [4,5]. Cells within these tissues are also more sensitive to various types of regulated (or programmed) cell death (RCD). RCD helps to aid in balancing cellular survival and demise, often in concert with tissue renewal processes, to promote tissue maintenance and attenuate tissue damage [6–9]. Many adaptive cellular stress responses impinge on these cell death pathways. Thus, RCD is a critical retort to irreversible cellular damage and must be precisely modulated to ensure proper epithelial tissue architecture and function in response to external or internal threats.

RCD and epithelial cell demise are often driven by the caspase-dependent apoptosis signaling pathway [9,10] and is governed by a balance of positive and negative inputs. Cellular stresses, such as reactive oxygen species (ROS)-induced oxidative stress, can lead to irreversible damage, intrinsic activation of apoptosis, and execution of RCD. Importantly, cells undergoing autonomous RCD will subsequently influence the local extracellular microenvironment by stimulating a local proinflammatory response (such as promoting the accumulation of extracellular ROS) [11]. These changes in the extracellular microenvironment can perturb stress responses and homeostasis in "neighbor" cells, leading to a propagation of cell death through secondary spreading of apoptosis-induced RCD (i.e., cell death promoting more cell death [12,13]). Changes in the extracellular microenvironment of epithelial tissues will also have a profound impact on the basement membrane, the extracellular matrix (ECM) that supplies a structural (cell attachment) and biochemical (signaling) platform for surrounding cells [14]. To this end, ECM–cell interactions have a critical impact on cell death regulation. Balancing cellular demise and survival, likely in coordination with the ECM, thus constitutes a key feature of resilience mechanisms that underlie the control of epithelial tissue damage.

The adult intestinal midgut of the arthropod *Drosophila melanogaster* provides an invaluable genetic model to explore ancestral resilience mechanisms of epithelial tissues. A simple (monolayer) barrier epithelium, the *Drosophila* midgut employs conserved stress responses to combat extrinsic (such as pathogens) or intrinsic (such as aging) threats, often through the coordination of cell death and tissue renewal. Our previous work identified Diedel, a protein secreted from peripheral tissues, as a potent antagonist of apoptosis-induced RCD in the intestinal midgut epithelium during aging, highlighting the integration of autonomous, local, and systemic responses in the control of cell death that shapes tissue damage [15]. Here, we show that Diedel is actually a ligand for RGD-binding Integrins, transmembrane receptors of epithelial cells that link the ECM with the cytoskeleton. Diedel thus shapes epithelial cell death by regulating cell–ECM interactions and cell attachment to the basement membrane. We also exploited Diedel function, as an in vivo genetic model, to explore ECM- and cell death–centric resilience mechanisms. Consequently, we uncovered a role for membrane-associated Catalase in preserving or repairing integrin–ECM interactions to maintain epithelial integrity. Intracellular Catalase, the ancient enzyme that is essential for quenching the ROS hydrogen peroxide, can relocalize to the extracellular membrane and limit ECM damage induced by the amplification of extracellular ROS after stress. This mechanism likely aids in maintaining epithelial structure and limits cell death spreading within the tissue. Membrane-associated Catalase can thus reduce the negative impact of adaptive cellular stress responses to limit epithelial tissue damage through preserving ECM–Integrin interactions and balancing cellular demise and survival.

## Results

### Diedel localizes at the basement membrane of the *Drosophila* midgut and shapes epithelial structure

We previously identified Diedel as a potent antagonist (inhibitor) of apoptosis-induced RCD both in vivo (in the adult *Drosophila* intestinal [midgut] barrier epithelium) and in vitro [15]. Diedel-dependent control of cell death significantly impacts both intestinal tissue aging and organismal lifespan [15] and possibly influences pathogen (virus)-mediated host defense mechanisms within the intestine [15,16]. As a secreted protein from peripheral (non-gastrointestinal) tissues (mainly produced and secreted from fat body/adipose tissue), Diedel represents a relatively unique molecule that can, likely, inhibit cell death extracellularly (adipose-to-midgut; [15]). Thus, we wanted to identify Diedel target proteins/receptors that could shape apoptosis and RCD in the *Drosophila* midgut epithelium. To this end, we first generated Diedel mutant fly lines using CRISPR-Cas9 genome engineering. Using this technology, 2 mutant lines were selected and characterized (S1A–S1D Fig). Mutant *die^{A1}* contains a single nucleotide (I) deletion and (II) mismatch that leads to a frameshift in translation (S1A and S1B Fig). This mutant is viable, generating homozygote adults and fertile females, but sterile males (S1C and S1D Fig). A second line (*die^{A2}*) contains a 5-nucleotide deletion that also leads to a frameshift in translation (S1A and S1B Fig), but these flies are not able produce viable homozygote adults due to lethality during development (during eclosion; S1C and S1D Fig). *die^{A1}*/*die^{A2}* trans-heterozygotes are viable, and additional analysis has revealed that the deletion in *die^{A2}* likely impacts expression of a neighboring gene (CG2310), which influences developmental viability [16].

We next used these mutant lines to explore Diedel's role in shaping midgut epithelial architecture and integrity. This adult midgut epithelial layer in *Drosophila* mainly contains 4 types of cells: intestinal stem cells (ISCs), enterocytes (ECs), secretory enteroendocrine cells (EEs), and enteroblasts (EBs, a postmitotic cell that can differentiate as an EC). The presence of ISCs promotes midgut regeneration of functional enterocytes after damage and enterocyte cell death [7,17–19]. Enterocytes, which are large, polyploid cells that project microvilli into the lumen, drive both nutrient absorptive functions and defense mechanisms. Toward the basement membrane (opposite the lumen), the epithelium is surrounded by visceral muscle exposed to circulating hemolymph (arthropod blood) [20]. Eliminating Diedel (*die^{A1}*/*die^{A1}* or *die^{A1}*/*die^{A2}*) leads to a defect in the adult midgut epithelium characterized by enterocyte detachment from the basement membrane (Fig 1A and 1B, additionally characterized in S1E Fig). We also observed a defect in visceral muscle fusion/attachment associated with the midgut and hindgut (Fig 1C). Attenuating Diedel secretion from adipose (fat body) utilizing Diedel^{RNAi} (CGGal4>UAS-Die^{RNAi}) also promotes enterocyte detachment from the basement membrane (Fig 1D and 1E). Diedel is inducible (actively secreted) in response to external or internal (such as aging) threats in order to regulate tissue damage [15,16,21–23], and these data show that Diedel is likely also required for the proper development and maintenance of midgut epithelial structure.

To further highlight a role for Diedel at the basement membrane, we generated transgenic flies that express a V5-tagged Diedel under the control of a 1,000-base pair region of the Diedel promotor (Die^{P}-DieV5, inserted as an additional copy). Immunostaining confirmed that Diedel protein is present in the fat body/adipose (S1F Fig) but also localizes at the basement membrane of the *Drosophila* midgut epithelium (Fig 1F). To confirm this, we assessed Diedel localization (utilizing Die^{P}-DieV5 flies) within a genetic background that endogenously expresses a multi-tag-GFP Laminin B1 (LanB1; [24]). The basal lamina is the layer of the ECM that is in direct contact with epithelial cells; it is formed by a self-assembling network of

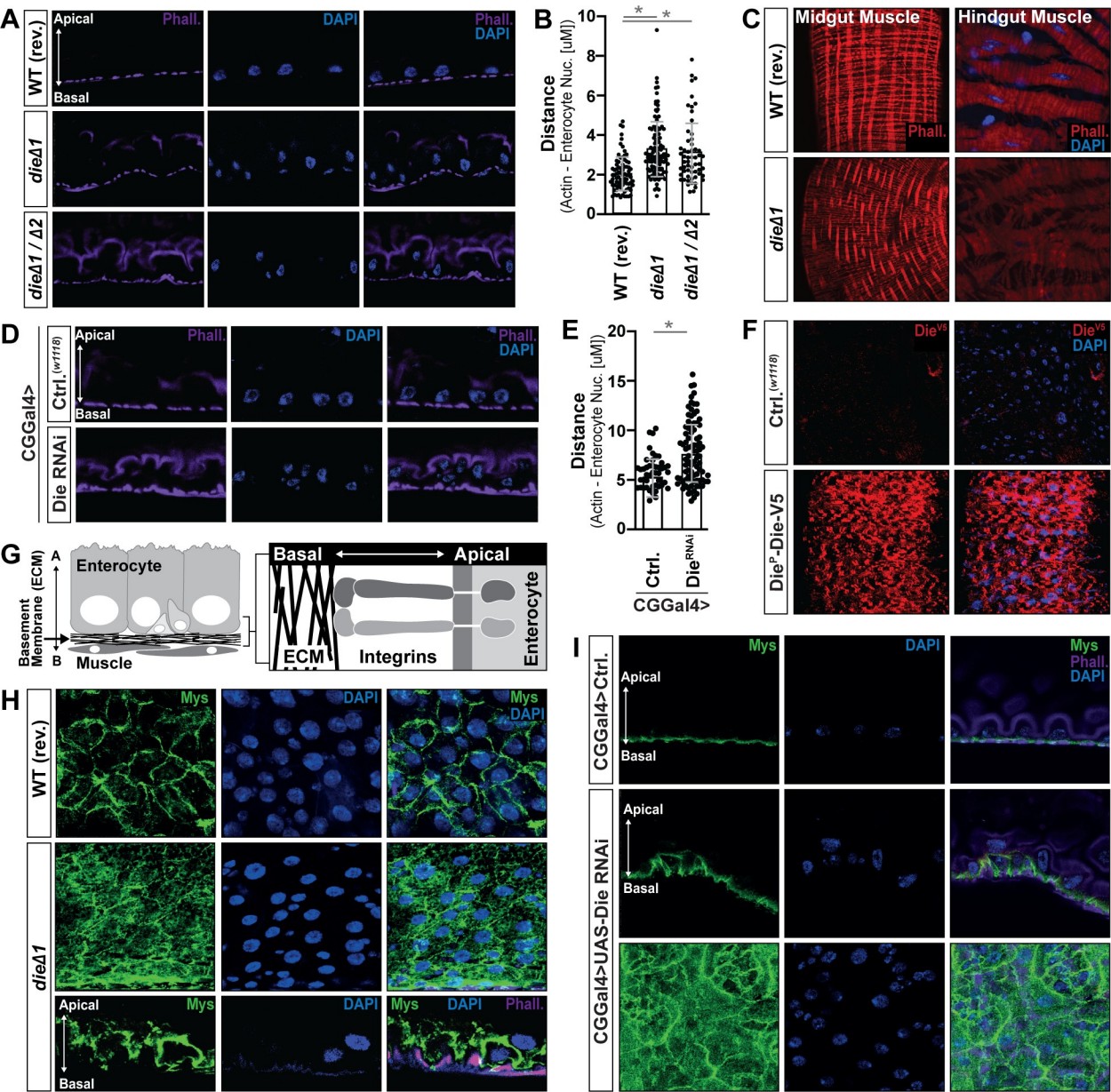

**Fig 1. Diedel is required for enterocyte attachment to the basement membrane.** (A) Dissected *Drosophila* posterior midgut cross sections from control (WT, revertant [rev.]), Diedel mutant homozygote (*dieΔ1/dieΔ1*), and Diedel mutant trans-heterozygote (*dieΔ1/dieΔ2*) flies; stained with Phalloidin (Phall.; purple) and DAPI (blue). (B) Distance quantification between posterior midgut enterocyte nuclei (identified as large nuclei; DAPI; blue) and visceral muscle (Phalloidin) from above genotypes; WT, revertant [rev.] n = 80; *dieΔ1/dieΔ1*, n = 119; *dieΔ1/dieΔ2*, n = 58. (C) Phalloidin staining (Phall.; red) of midgut and hindgut visceral muscle dissected from WT, *dieΔ1/dieΔ1*, and *dieΔ1/dieΔ2* flies. (D) Dissected posterior midgut cross sections from control flies (*w1118*; CGGal4) and flies with fat body attenuation of Diedel (*w1118*; CGGal4 / UAS-Die RNAi); stained with Phalloidin (Phall.; purple) and DAPI (blue). (E) Distance quantification between posterior midgut enterocyte nuclei (identified as large nuclei; DAPI; blue) and visceral muscle (Phalloidin) from above genotypes; *w1118*; CGGal4, n = 45; *w1118*; CGGal4 / UAS-Die RNAi, n = 85. (F) Die-V5 localization in dissected *Drosophila* midguts from controls (negative control; *w1118*) and *w1118*; Die$^P$-Die-V5 / Die$^P$-Die-V5 transgenic flies; stained with anti-V5 antibody (Die-V5; red) and DAPI (blue). (G) Schematic of *Drosophila* midgut organization (basement membrane). (H) Integrin Mys immunostaining of dissected posterior midguts from control (WT, revertant [rev.]) and Diedel mutant (*dieΔ1/dieΔ1*) flies; bottom panel represents cross-section (apical–basal polarity is highlighted with arrow); stained with anti-Mys (green), Phalloidin (Phall.; purple), and/or DAPI (blue).(I) Integrin Mys immunostaining of dissected posterior midguts from control flies (*w1118*; CGGal4) and flies with fat body attenuation of Diedel (*w1118*; CGGal4 / UAS-Die RNAi); top panels represent cross-sections (apical–basal polarity is highlighted with arrow); stained with anti-Mys (green), Phalloidin (Phall.; purple), and/or DAPI (blue). Data information: Data in panels B and E are presented as mean ± SD, n = 45–109. *P ≤ 0.05 (Student *t* test). The data underlying the graphs shown in Fig 1B and 1E can be found in S1 Data.

Laminin heterotrimers consisting of an alpha, beta, and gamma subunit [25,26]. Immunostaining confirmed colocalization of DieV5 with LanB1 at the midgut epithelium basement membrane (S1G Fig).

In totality, these data suggest that secreted Diedel is necessary to maintain midgut epithelial integrity by enabling enterocyte (and potentially muscle cell) attachment to the basement membrane and that Diedel maybe is a critical component of the ECM associated with the basement membrane.

## Diedel is an Integrin ligand, with a functional RGD domain, required for attaching epithelial cells to the ECM

The basement membrane of the midgut epithelium is comprised of various ECM components, including laminins, collagens, and proteoglycans [27–31]. Integrins are highly conserved transmembrane receptors, functioning as heterodimers of alpha and beta chains, which molecularly link the ECM to cell membranes through interactions with cytoskeletons (Fig 1G) [28,32]. These receptors are thus essential for epithelial cell attachment to the basement membrane. *Drosophila* enterocytes express 1 main integrin beta chain (Mys; Myospheroid) and 4 different alpha chains (Mew, If, Scb, and ItgaPS4) [33]. Focusing on Mys, Integrins are normally localized at the basal side (toward the basement membrane [ECM]) of enterocytes (Fig 1H and S2A Fig). However, when Diedel is eliminated ($die^{A1}/die^{A1}$) or cell-specifically attenuated (CGGal4>UAS-Die$^{RNAi}$), Mys is delocalized from the enterocyte–ECM boundary (Fig 1H and 1I), suggesting that Diedel is necessary for Mys basal localization and maintaining apical–basal polarity (related to Integrin–ECM interactions) of enterocytes. Similar results were obtained utilizing a temperature sensitive driver to attenuate Diedel cell-specifically in the adult fly (PplGal4,TubGal80$^{ts}$> UAS-Die$^{RNAi}$; S2B Fig).

Furthermore, utilizing Die$^P$-DieV5 transgenic flies in a $die^{A1}/die^{A1}$ mutant background, we found that Die-V5 binds to the basal side of enterocytes and colocalizes with the Integrin Mys (Fig 2A). This transgenic construct also shows that reexpressing Diedel (DieV5) can rescue the loss of Mys polarity in Diedel mutant flies (Fig 2A) and highlights that these Integrin delocalization phenotypes are, indeed, influenced by Diedel function.

Integrin heterodimer receptors bind to specific Integrin ligands [31,34,35]. Integrin ligands are diverse but are required, in part, for maintaining functional ECM–Integrin–epithelial cell connectivity (i.e., maintaining cell attachment to the basement membrane). Integrin ligands are thus often components of the ECM. A motif analysis of the Diedel protein sequence revealed an RGD (Arg-Gly-Asp) domain within an accessible epitope in the Diedel protein (Fig 2B; [36]). In *Drosophila*, ECM proteins with an RGD domain have been shown to bind specifically to the Integrin heterodimer Myospheroid-Inflated (Mys-If; [37–40]). To determine if Diedel is an Integrin ligand, we turned to an in vitro "cell spreading" assay utilizing *Drosophila* S2R+ cells [39,41], as we had previously used these cells to inform on Diedel function [15]. First, we constructed in vitro expression plasmids containing either wild-type V5-tagged Diedel (Die$^{RGD}$-V5) or a Diedel protein where the RGD sequence was replaced by an RGE (Arg-Gly-Glu) sequence (1 amino acid substitution, Die$^{RGE}$-V5). This substitution is commonly used to inhibit the interaction between RGD domain ligands and Integrins [42]. Transfection of expression plasmids containing tagged RGD or RGE Diedel proteins in S2R+ cells revealed that these proteins are readily secreted into media (Fig 2C). Using conditioned media enriched with Die$^{RGD}$-V5 or Die$^{RGE}$-V5, we found that S2R+ cells transfected with Integrin Inflated are able to "spread" on a coated cover slip in the presence of Die$^{RGD}$-V5 conditioned media, indicative of cell attachment and cytoskeletal rearrangement (Fig 2D and 2E) However, replacing the RGD domain with RGE in abolishes this "spreading" (Die$^{RGE}$-V5; Fig 2D and 2E).

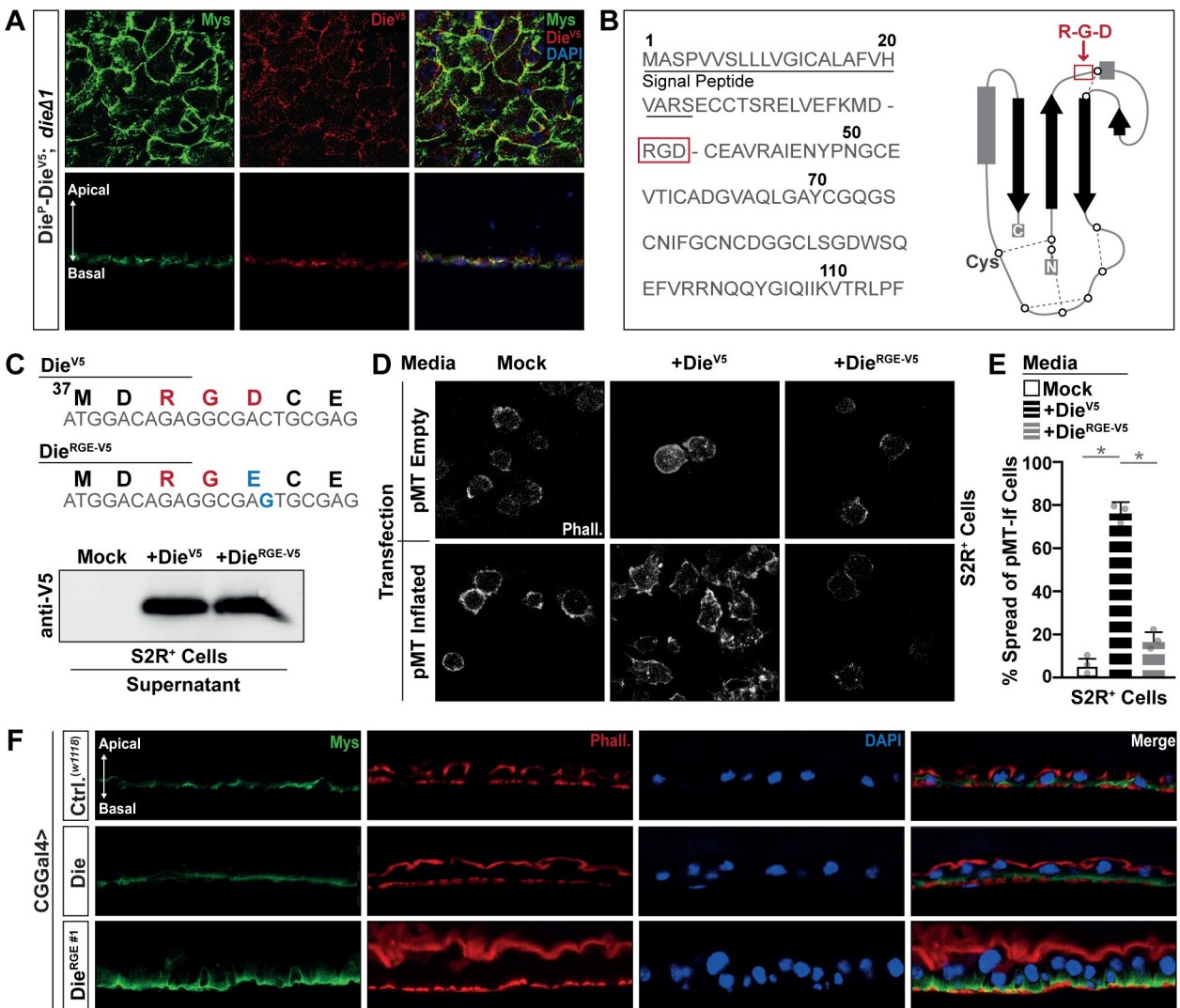

**Fig 2. Diedel is a ligand for RGD-binding Integrins.** (A) Integrin Mys and anti-V5 immunostaining of dissected posterior midguts from *w1118*; Die^P-Die-V5 / Die^P-Die-V5; *dieΔ1/dieΔ1* transgenic flies; bottom panel represents cross-section (apical–basal polarity is highlighted with arrow); stained with anti-Mys (green), anti-V5 antibody (Die-V5; red), and DAPI (blue). (B) Diedel protein sequence and topology with RGD domain highlight in red. (C) Die-V5 and DieRGE-V5 protein sequences and nucleotide sequence alignment; nucleotide and amino acid substitution highlighted in blue (top panel). Bottom panel; western blot with anti-V5 antibody of conditioned media (supernatant) from S2R+ cells either nontransfected (Mock), transfected with Die-V5, or transfected with DieRGE-V5. (D) Transfection of S2R+ cells (control; pmT empty) with Inflated Integrin (pMT Inflated), plated on conditioned media (supernatant) containing Mock (control), Die-V5, or DieRGE-V5; cells stained with Phalloidin (Phall.; white). (E) Quantification of "cell spreading" assay in S2R+ cells plated on various conditioned medias. The percentage of "spread" cells is shown, and bars indicate mean ± SD from *n* = 3 independent experiments (approximately 100 cells counted in each experiment). *P ≤ 0.05 (Student *t* test). (F) Integrin Mys immunostaining of dissected posterior midguts from control flies (*w1118*; CGGal4) and flies with fat body expression of either wild-type Diedel (*w1118*; CGGal4 / UAS-Die) or Diedel RGE (*w1118*; CGGal4 / UAS-Die^RGE#1), stained with anti-Mys (green), Phalloidin (Phall.; red), and/or DAPI (blue). The data underlying the graphs shown in Fig 2E can be found in S1 Data.

To confirm the function of this Diedel RGD domain in vivo, we generated transgenic flies that express UAS-linked Diedel containing the RGD-to-RGE substitution used in vitro (UAS-Die^RGE). Overexpression of Diedel^RGE from the fat body (CGGal4>UAS-Die^RGE) induced disorganization of the midgut epithelium characterized by enterocyte detachment and integrin (Mys) delocaliztion (Fig 2F, and additional transgenic line is characterized in S2C Fig). This phenotype is characteristic of Diedel mutants, suggesting that Diedel^RGE can competitively antagonize normal Diedel (Diedel^RGD) function.

Taken together, these data show that Diedel is an ECM protein and an Integrin ligand, with a functional RGD domain. Diedel–Integrin–ECM interactions are thus required to maintain midgut epithelial tissue integrity through regulating epithelial cell attachment to the basement membrane.

## Diedel–Integrin–ECM interactions antagonize cell death spreading by promoting cell survival

Next, we wanted to explore the connection between Diedel, the ECM, and Integrins in governing cell death in the midgut epithelium. Our previous work highlighted the ability of Diedel overexpression to robustly inhibit apoptosis-induced RCD in vitro and in vivo [15]. Apoptosis-induced RCD is highly conserved in *Drosophila* [9,10], as various apoptotic responses are mediated by initiator caspases (Dronc; Cas-9-like) [43] and effector caspases (DrICE [44] and Dcp-1 [45]; Cas-3-like), as well as a family of IAP (Inhibitor of apoptosis; DIAP1) antagonists (Hid [46], Reaper (Rpr) [47], and Grim [48]) governed by Jun-N-terminal kinase (JNK) activity (S3A Fig). In the *Drosophila* midgut, enterocyte cell death initiates tissue renewal (stem cell division/differentiation) in response to intrinsic and extrinsic cues. Our previous analysis of Diedel function in the control of apoptosis exclusively utilized endpoint analysis of RCD (measuring effector caspase activation and DNA fragmentation) [15]. Additionally, we were unable to find evidence that Diedel can regulate canonical intrinsic cell death responses during fly development (examples provided in S3B–S3F Fig). Thus, we hypothesized that Diedel is not able to inhibit apoptosis in cells that have already initiated regulated cell death. Instead, due to its presence in the ECM, Diedel may inhibit the propagation of cell death (cell death spreading via the extracellular environment) to neighboring cells. To test this, we exploited an in vitro trans-well cell culture system where damaged/dying cells are separated from undamaged cells via a 3-μm mesh membrane that permits small molecule transfer. First, we established that Diedel (using conditioned media with Die-V5) can bind membranes of Drosophila S2R+ cells after UV treatment (a stimuli that induces apoptosis; S2A Fig), again highlighting Diedel's ability to associate with the ECM in cell culture (Fig 3A). Utilizing the trans-well culture system (Fig 3B), we found that UV-induced cell death can "spread" to non-UV-treated "neighbor" cells to induce RCD (assayed by Annexin V staining), and the secreted Diedel (conditioned media with Die-V5) can block these cell death responses transmitted via the extracellular microenvironment (Fig 3C). Furthermore, while Diedel cannot inhibit cell death that is stimulated directly by genetic induction of intrinsic caspase activation (through RNAi-mediated inhibition of DIAP [49]), it can potently block the spreading of cell death responses initiated by these irreversibly damaged (dying) apoptotic cells (Fig 3D).

These data show that Diedel likely limits the propagation of cell death, preventing secondary spreading of apoptosis-induced RCD, and highlights a putative extrinsic mechanism to control RCD in insects. Since we found that Diedel is an ECM protein and can bind Integrins, we hypothesized that dying cells induce remodeling of the extracellular microenvironment, leading to the loss of ECM–Integrin interactions, epithelial cell detachment, and cell death [50]. To begin to address this complex hypothesis, we first explored the ability of Diedel to dictate Talin localization in the midgut epithelium in vivo. Functional interactions between Integrins and the ECM can be assessed by the recruitment of Talin at the cellular membranes, as Talin (along with other cofactors) is essential for ECM–Integrin–cytoskeleton attachments (Fig 3E) [51,52]. However, when Integrins are unligated (unable to interact with the basement membrane ECM), this can lead to mislocalization of Talin, which has been linked to cell detachment–mediated cell death [53]. Using immunostaining, we found that when Diedel is eliminated (*die*^{A1}/*die*^{A1} or *die*^{A1}/*die*^{A2}) or cell-specifically attenuated (CGGal4>UAS-Die^{RNAi}),

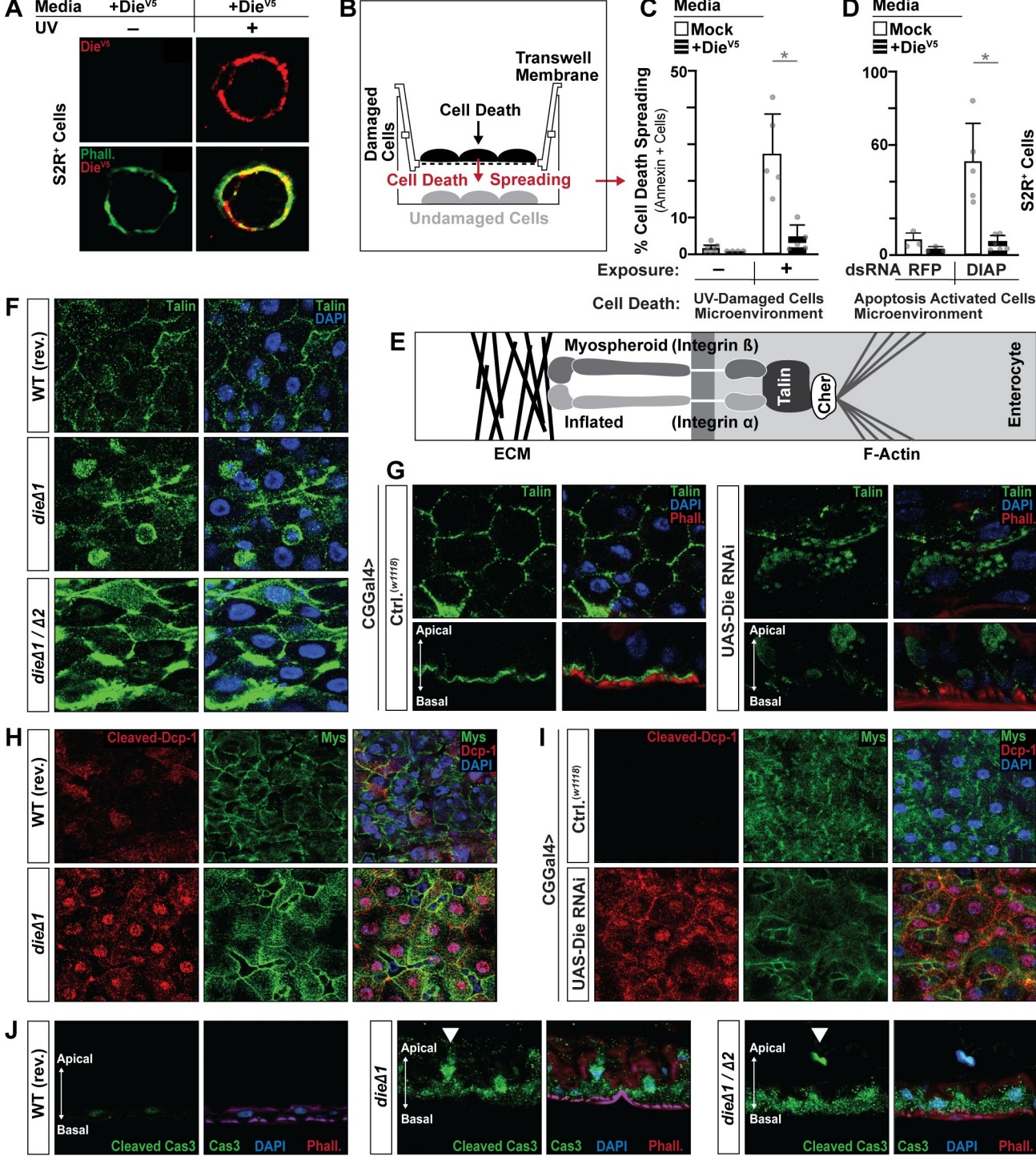

**Fig 3. Diedel prevents cell death spreading initiated by loss of Integrin–ECM interactions.** (A) Anti-V5 immunostaining of S2R+ cells before and after UV exposure (100 mJ/cm²); cells treated with Die-V5 conditioned media (supernatant); stained with anti-V5 antibody (Die-V5; red) and Phalloidin (Phall.; green). (B) Schematic representing coculture system to study cell death spreading in vitro. (B) UV-induced (100 mJ/cm²; +/− exposure) cell death spreading; quantified by percentage of Annexin V–positive cells. S2R+ cells treated with conditioned media (supernatant) containing Mock (control) or Die-V5. (C) Caspase (apoptosis)-induced (dsRNA targeting DIAP; dsRNA RFP as control) cell death spreading; quantified by percentage of Annexin V–positive cells. S2R+ cells treated with conditioned media (supernatant) containing Mock (control) or Die-V5. (D) Schematic of the ECM–Integrin–Talin complex; Cher (Cheerio) is an additional adaptor protein. (E) Talin immunostaining of dissected posterior midguts from control (WT, revertant [rev.]) and Diedel mutant (*dieΔ1/dieΔ1* and *dieΔ1/dieΔ2*) flies; stained with anti-Talin (green) and DAPI (blue). (F) Talin immunostaining of dissected posterior midgut from control flies (*w1118*; CGGal4) and flies with fat body attenuation of Diedel (*w1118*; CGGal4 / UAS-Die RNAi); stained with anti-

Talin (green), DAPI (blue), and Phalloidin (Phall.; red); bottom panel represents cross-section (apical–basal polarity is highlighted with arrow). (G) Integrin Mys and Caspase (cleaved Dcp-1) immunostaining of dissected posterior midguts from control (WT, revertant [rev.]) and Diedel mutant (*dieΔ1*/*dieΔ1*) flies; stained with anti-Mys (green), anti-Dcp-1 (Cleaved Dcp-1; red), and DAPI (blue). (H) Integrin Mys and Caspase (cleaved Dcp-1) immunostaining of dissected posterior midguts from control flies (*w1118*; CGGal4) and flies with fat body attenuation of Diedel (*w1118*; CGGal4 / UAS-Die RNAi); stained with anti-Mys (green), anti-Dcp-1 (Cleaved Dcp-1; red), and DAPI (blue). (I) Posterior midgut cross sections (apical–basal polarity is highlighted with arrow) from control (WT, revertant [rev.]) and Diedel mutant (*dieΔ1*/*dieΔ1* and *dieΔ1*/*dieΔ2*) flies; stained with anti-cleaved Cas3 (green), DAPI (blue), and Phalloidin (Phall.; red). Strong staining of Cas3 (revealing cells detaching from the basement membrane) are highlighted by solid white arrowhead). Data information: Data in panels C and D are presented as mean ± SD, *n* = 3–5. *$P \leq 0.05$ (Student *t* test). The data underlying the graphs shown in Fig 3C and 3D can be found in S1 Data.

Talin is mislocalized to the cytoplasm of enterocytes (losing apical–basal polarity; Fig 3G; ctrl.) and can be also found in the nucleus (Fig 3F and 3G).

Despite this significant loss of ECM–Integrin–cytoskeleton attachments, Diedel mutant midguts maintain barrier function and overall midgut cellular architecture (S3G Fig; assayed by midgut permeability [54,55]), while also displaying heightened renewal responses (ISC proliferation; S3H Fig). Expressly, it appears that not all epithelial cells execute apoptosis-induced RCD, something we had also discovered previously [15], despite significant increases in cell death markers and the breakdown in cell polarity (S3I Fig). Thus, we wanted to further explore extracellular-mediated cell death linked to ECM–Integrin function. Utilizing immunostaining in the *Drosophila* midgut, we found that loss of Integrin–ECM interactions, established by Diedel loss-of-function, is associated with the activation (cleavage) of the effector Caspase Dcp-1 in midgut enterocytes (Fig 3H and 3I). Cleaved Dcp-1 can be found enterocyte nuclei, suggesting that active Dcp-1 is translocated from the cytoplasm into the nucleus during progression through apoptosis, similar to Caspase-3 in mammalian cells [56]. Although, loss of nuclear membrane integrity leading to this phenotype cannot be excluded. Moreover, we also found cleaved Dcp-1 at the cellular plasma membrane (likely intracellular membranes, as Dcp-1 is not secreted). To this end, a focused analysis revealed that cleaved Dcp-1 does not colocalize at the basement membrane (with Mys) in Diedel mutant epithelial cells (S4A Fig) and instead appears to localize to enterocyte cell junctions (at intracellular enterocyte membranes). To confirm this, we found that cleaved Dcp-1 colocalizes with Dlg1 (Discs large 1) at enterocyte cell–cell junctions (S4B Fig). Dlg1 is a marker for septate junctions of epithelial cells, is required for enterocyte cell polarity [57], and is associated with cell death induced by cell detachment (anoikis) [58]. Recruitment of effector Caspases to cell junctions could limit the spread of cell death within epithelial tissues, attenuating or delaying the activation of terminal Caspases that lead to irreversible cellular demise as part of resistance mechanisms to limit tissue damage. Our data suggest that cell–cell interactions may be involved to prevent the execution of the apoptosis-induced cell death [59] by recruiting/retaining effector caspases at septate junctions. A similar mechanism was identified in vitro using a human colon epithelial cell line, where Cadherins (important for cell–cell adherens junctions) are involved in cell death protection by preventing activation/execution of the Caspase pathway [58]. Accordingly, we uncovered that in Diedel mutants (*die$^{Δ1}$*/*die$^{Δ1}$* or *die$^{Δ1}$*/*die$^{Δ2}$*), strong activation (cleavage) of Cas-3 in enterocytes (an indication of dying cells) is mainly revealed in epithelial cells that are detached (shedding) from the basement membrane and shifting apically toward the lumen (Fig 3J).

To additionally characterize the role of Diedel–Integrin–ECM interactions in cell death and cell death spreading, we generated chimeric Integrin receptors that express the extracellular domain of Murine CD8 alpha conjugated to the transmembrane and intracellular domain of the *Drosophila* Integrin Mys (S5A Fig). Expression of these types of chimeric receptors were shown to mimic loss of Integrin–ECM interactions and can lead to cell death [60,61]. However, using this biochemical model, we were not able to show any direct cell death induction in *Drosophila* S2R+ cells (S5A–S5C Fig), suggesting at least that Diedel-and-Integrin–mediated

loss of cell attachment to the basement membrane may not initiate cell death spreading or directly activate RCD. Alternatively, Integrin–membrane interactions are well known to preserve cell viability in response to stress through the activation of Raf / MEK (mitogen-activated ERK kinase) / ERK (extracellular signal–related kinase) signaling pathway (MAPK; [62,63]). Activation of the MAPK pathway in *Drosophila* was shown to specifically inhibit caspase activation [64]. Utilizing a temperature sensitive driver to attenuate or overexpress Diedel cell-specifically in the adult fly (PplGal4,TubGal80^ts), we assessed MAPK pathway activation in the midgut epithelium using an antibody directed against phospho-ERK (immunostaining). Diedel overexpression (PplGal4,TubGal80^ts>UAS-Die) leads to enhanced phospho-ERK immunostaining in midgut enterocytes (large nuclei; S5D Fig). This suggests that Diedel–Integrin–ECM interactions likely shape the balance of cell death and survival in epithelial tissues. Coupled with our previous work showing that Diedel is also essential to properly regulate midgut epithelial regeneration/renewal after stress [15], these findings highlight that Diedel–Integrin–ECM interactions promote resilience of epithelial tissues, i.e., the ability of epithelial tissues to endure stress and recover.

In totality, by exploring Diedel function, our data show that Diedel, an Integrin ligand, is critical to promote resilience mechanisms that support the protection of epithelial tissues and poise cell death and survival.

## Membrane-associated Catalase is necessary to prevent cell death spreading associated with Diedel–Integrin–ECM interactions

The ECM is a network of macromolecules that both provides cellular scaffolding and initiates critical biochemical and mechano-sensing cues that determine various cellular dynamics. This network is in perpetual transformation through the action of enzymes that build or disassemble the matrix and allow for cell migration, proliferation, or shedding, as well as directing signaling pathway activity, in order to coordinately promote epithelial tissue renewal/maintenance, and balance cellular demise and survival. To this end, we exploited Diedel's role in cell survival, and its function as an Integrin ligand associated with extracellular membranes, to better characterize the response of resilient epithelial cells to stress-induced changes in the extracellular environment. First, we utilized S2R+ cells and unbiased mass spectrometry to explore the more complex interactome associated (directly or indirectly) with Diedel–Integrin–ECM interactions at cell membranes. UV treatment was used to induce Diedel localization (conditioned media with Die-V5) to cellular membranes of S2R+ cells followed by crosslinking (with paraformaldehyde), and Die-V5 was subsequently pulled down with putative membrane/ECM-associated proteins and separated through gel electrophoresis (Fig 4A). Crosslinking will create covalent bonds between proteins, driving the pulldown of putative direct interacting targets, but also indirectly associated proteins present at membranes (based on proximity) when Diedel–Integrin–ECM interactions are enriched. Mass spectrometry protein identification identified that the antioxidant enzyme Catalase (Cat) and Inositol-3-phosphate synthase (Inos) are associated at membranes when Diedel is also enriched (Fig 4A). To confirm these associations, since crosslinking likely creates large protein complexes that can be difficult to gel-separate, we repeated this pulldown, but instead of gel separation, we analyzed the entire elution fraction (again using mass spectrometry; Fig 4B and 4C, S1 Table). These data confirmed the presence of Catalase and Inos at cellular membranes enriched for Diedel–Integrin–ECM interactions and identified other putative membrane-associated proteins linked to antioxidant function (CaBP1; [65]) or Integrin function (Rack1; [66,67]).

We decided to focus on Catalase, since metazoan Catalase is normally found within the cytoplasm in order to decompose intracellular hydrogen peroxide (ROS), and identifying

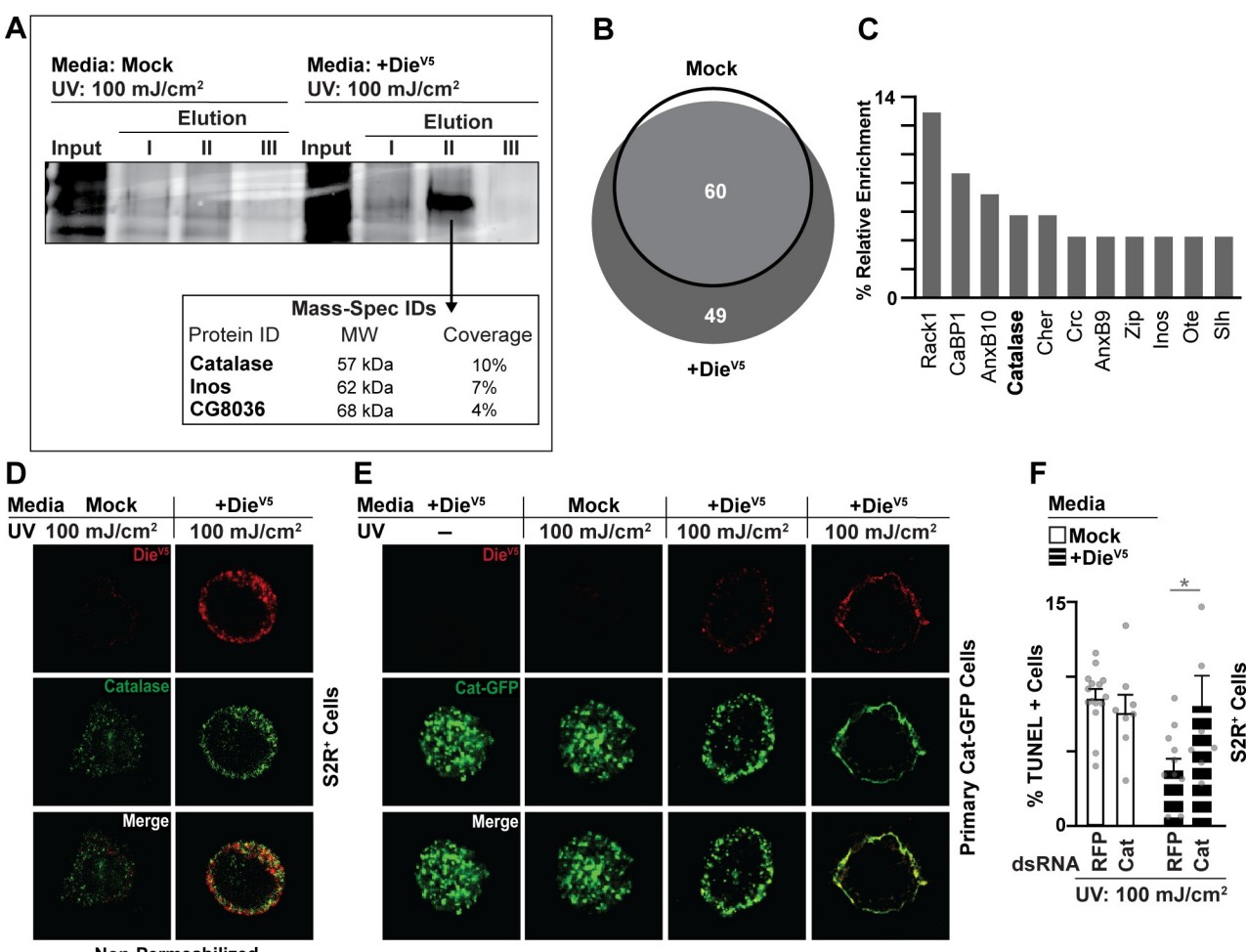

**Fig 4. Catalase associates with cellular membranes/ECM after stress.** (A) Mass spectrometry of gel-based pulldown analysis. Protein identification (gel elutions provided) after Die-V5 pulldown from S2R+ cells treated with control (Mock) or with Die-V5 conditioned media (supernatant); after stress (UV exposure; 100 mJ/cm$^2$). Table represents Mass-Spec protein identification with percent protein coverage. (B) Mass spectrometry of gel-free pulldown analysis (Diedel-interacting proteins). Venn diagrams showing overlap of proteins after Die-V5 pulldown from S2R+ cells treated with control (Mock) or with Die-V5 conditioned media (supernatant); after stress (UV exposure; 100 mJ/cm$^2$). The threshold for proteins included in the analysis was at least 2 different fragments identified/sequenced. (C) Histogram plotting percentage of protein enrichment in S2R+ cells treated with Die-V5 conditioned media after UV exposure. The analysis included only cytoplasmic and membrane proteins with at least 2 different fragments identified/sequenced. (D) Catalase and anti-V5 immunostaining of S2R+ cells after UV exposure (100 mJ/cm$^2$); cells treated with Die-V5 conditioned media (supernatant) or Mock (control); stained with anti-V5 antibody (Die-V5; red) and anti-Catalase (green). Cells were not permeabilized (nonpermeabilized) with detergent in order to visualize membrane/ECM proteins. (E) Catalase localization in primary cells expressing endogenous catalase tagged with GFP (green) after UV exposure (100 mJ/cm$^2$); cells treated with Die-V5 conditioned media (supernatant) or Mock (control); stained with anti-V5 antibody (Die-V5; red). (F) UV-induced (100 mJ/cm$^2$; +/− exposure) cell death quantification by percentage of TUNEL-positive cells. S2R+ cells were treated with dsRNA control (RFP) or dsRNA Catalase (cat), then exposed to UV (100 mJ/cm$^2$) and treated with Die-V5 conditioned media (supernatant) or Mock (control). Data information: Data in panel F are presented as mean ± SD, *n* = 8–14. *$P \leq 0.05$ (Student *t* test). The data underlying the graphs shown in Fig 4B can be found in S1 Table, and the data underlying the graphs shown in Fig 4C and 4F can be found in S1 Data.

Catalase at cellular membranes was unexpected. Although, extracellular Catalase has been linked with transformation of cancer cells [68] and characterization of purified Catalase from *Drosophila* suggested a significant fraction may be membrane bound [69]. We confirmed in vitro that Catalase colocalize at cellular membranes of Diedel-positive cells after stress (Fig 4D). Using a Catalase antibody, Catalase can be uncovered intracellularly in S2R+ cells before and after stress (UV treatment; S6A Fig). However, in the absence of cell permeabilization

(sequestering antibodies outside of the cell), we found Catalase at cellular membranes after stress (Fig 4D). We confirmed these results, in vitro, utilizing primary embryonic *Drosophila* cells derived from flies containing an endogenously tagged (GFP) Catalase within the genome [70] (Cat-GFP; Fig 4E and S5B Fig). While Catalase is present at membranes enriched for Diedel–Integrin–ECM interactions, co-immunoprecipitations in native (non-crosslinking) conditions revealed that Diedel and Catalase do not directly interact (S5C Fig), and thus Catalase is likely part of a larger membrane interactome, associated with Integrin–ECM interactions, which may shape cell death and survival responses. To this end, we also found that Catalase is required for the ability of Diedel to prevent cell death in vitro (Fig 4F).

In *Drosophila*, dying epithelial cells (midgut enterocytes) increase ROS generation [11]. Thus, ROS (likely extracellular ROS such as hydrogen peroxide) may promote cell death spreading and ECM remodeling/degradation through oxidation to drive cell detachment from the basement membrane. It is probable that extracellular ROS generation needs to be tightly controlled to avoid cell death spreading from the dying cells within epithelial tissues. We hypothesized that membrane-associated Catalase could play a critical role in reestablishing Redox (ROS-mediated reduction/oxidation) balance to prevent cell death spreading, repair/preserve ECM–Integrin interactions, and promote tissue resilience after stress. To test this, we first confirmed that Catalase can localize to membranes in vivo. Using a Catalase antibody, we found extracellular Catalase in the midgut epithelium after DSS treatment/feeding (Fig 5A). DSS (a derivative of dextran) induces ROS, as well as alter the basement membrane and promote tissue damage in the *Drosophila* midgut [71]. Membrane-associated Catalase was visualized by utilizing nonpermeable conditions. To determine if membrane-associated Catalase can be found nonautonomously in response to epithelial cell death, we generated RFP-marked clones (using MARCM [mosaic analysis with a repressible cell marker] in the Cat-GFP genetic background) in the midgut expressing both the proapoptotic caspase initiator Reaper (UAS-Rpr; see also S3A Fig) and an inhibitor of apoptosis (UAS-P35) [72,73]. Coexpression of Reaper and the baculovirus P35 creates "undead" cells; cells initiating, but not completing, apoptosis [11,73–75]. This MARCM clonal analysis was performed within transgenic flies containing the endogenously tagged Catalase-GFP (S5B Fig). Indeed, immunostaining revealed Catalase relocalizing to membranes in cells adjacent ("neighbor") to "undead" midgut clones (Fig 5B), suggesting that membrane-associated Catalase may be critical in regulating cell death spreading in synergy with Integrin–ECM interactions.

We next wanted to assay Catalase-dependent changes in extracellular ROS in the midgut epithelium. Directly measuring extracellular ROS and H2O2 specifically in vivo is challenging, so we choose an indirect approach. Cysteine-rich proteins, such as Laminins, Secreted protein acidic and rich in cysteine (SPARC), and even Diedel, are abundant in the intestinal epithelium ECM. Thus, cysteine disulfide bond plays an important role in ECM protein–protein interactions and cohesion. The thiol group of cysteine is sensitive to ROS (or Redox balance) in the extracellular microenvironment. Exploiting a dimedone-based chemical probe, coupled with an antibody, to explore changes in thiol oxidation [76], we can indirectly measure extracellular ROS. Although dimedone is permeable, we performed secondary immunostaining in nonpermeable conditions to sequester antibodies extracellularly and thus visualize the oxidation status of extracellular thiol-cysteines. Using transgenic flies containing the endogenously tagged Cat-GFP (S5B Fig), we found that DSS treatment/feeding leads to increases oxidized cysteines at the ECM/membrane of enterocytes, which also correlates with Catalase localization to membranes (Fig 5C).

Furthermore, cotreatment/feeding with 3-amino-1,2,4-triazole (3-AT), an irreversible Catalase inhibitor [77], enhances DSS-mediated increases in extracellular oxidation of cysteines and promotes enterocyte detachment (Fig 5C and S5D Fig). This increase in staining likely

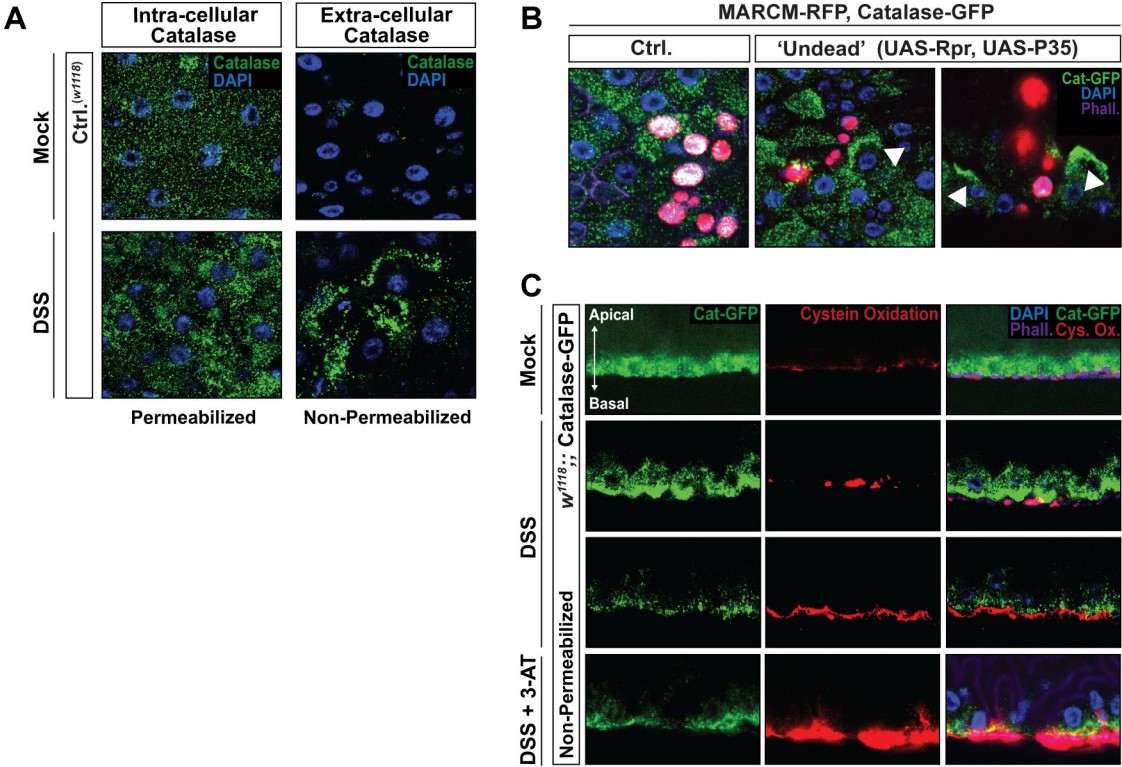

**Fig 5. Catalase relocalization to membranes is integrated with cell death spreading and extracellular ROS after epithelial tissue damage.** (A) Catalase immunostaining of dissected posterior midguts from control (*w1118*, Ctrl.) flies with DSS treatment/feeding (4% DSS) or mock treatment; stained with anti-Catalase (green) and DAPI (blue). Flies were fed (or Mock treated) DSS for 2 days. Immunostaining was performed in both permeabilized (to visualize intracellular Catalase) and nonpermeabilized (to visualize extracellular Catalase) conditions. (B) Cat-GFP immunostaining in dissected midguts from control (Ctrl.) MARCM clones and "Undead" MARCM clones 5 days after heat shock. Clones are marked with nuclear RFP (red), endogenously GFP-tagged Catalase (Catalase-GFP/+) are marked with GFP (green). White arrows highlight membrane localized Catalase in adjacent ("neighbor") cells of "undead" MARCM clones. (C) Catalase localization and membrane/ECM cysteine oxidation of dissected posterior midguts from *w1118*, Catalase-GFP / Catalase-GFP (endogenously tagged) flies; DSS treatment/feeding (4% DSS), or mock treatment, or cotreatment/feeding with a Catalase inhibitor (DSS + 3-AT); stained with anti-Catalase (green), anti-Cysteine (oxidized, Cys. Ox; red), and DAPI (blue). Immunostaining was performed in nonpermeabilized conditions to visualize membrane/ECM cysteine oxidation. Panels represent cross-sections (apical–basal polarity is highlighted with arrow). Flies were fed (or Mock treated) DSS or DSS + 3-AT for 2 days. ECM, extracellular matrix; MARCM, mosaic analysis with a repressible cell marker; ROS, reactive oxygen species.

represents, in part, changes in extracellular ROS and suggests that Catalase (putatively membrane-associated Catalase) dictates Redox balance within the ECM/membranes of epithelial tissues. Accordingly, Catalase inhibition attenuates the recovery of the midgut epithelium, as the presence of oxidized cysteines is still prevalent 48 hours posttreatment/feeding of DSS (Fig 6A). Even when Catalase function is inhibited, which drives increases in intracellular and extracellular ROS, Catalase still relocalizes at midgut enterocyte membranes, suggesting that these molecular changes underly resilience mechanisms in epithelial tissues. Membrane-associated Catalase is thus likely required to limit the burden of adaptive stress responses that generate extracellular ROS and subsequently limit tissue damage.

Disulfide bond formation is also critical to integrate Laminin to basement membrane and thus maintain ECM architecture and interactions with Integrins [78]. To confirm a role for Catalase in ECM–Laminin incorporation, we isolated *Drosophila* midgut ECM proteins by decellularizing the gut (S7A Fig) and resolving ECM/extracellular proteins on a gel. Treatment/feeding of the Catalase inhibitor 3-AT leads to a decrease of LanB2 incorporation into

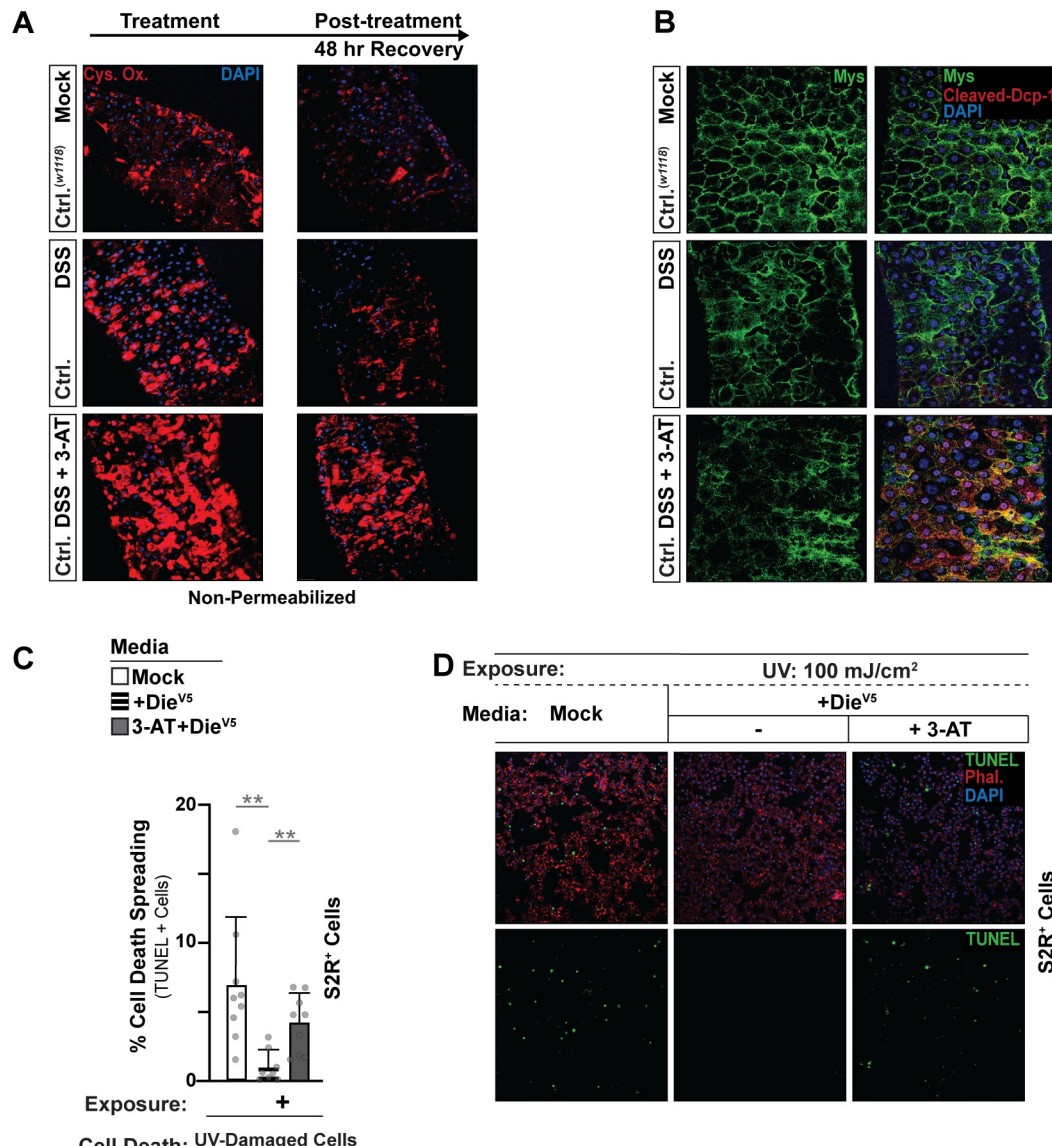

**Fig 6. Membrane-associated Catalase is necessary to promote ECM preservation / repair and prevent cell death spreading in concert with Diedel–Integrin–ECM interactions.** (A) Membrane/ECM cysteine oxidation of dissected posterior midguts from *w1118* (Ctrl.) flies; DSS treatment/feeding (4% DSS), or mock treatment, or cotreatment/feeding with a Catalase inhibitor (DSS + 3-AT), as well as 48 hours posttreatment (recovery after DSS feeding); stained with anti-Cysteine (oxidized, Cys. Ox; red) and DAPI (blue). Immunostaining was performed in nonpermeabilized conditions to visualize membrane/ECM cysteine oxidation. Flies were fed (or Mock treated) DSS or DSS + 3-AT for 2 days. (B) Integrin Mys and Caspase (cleaved Dcp-1) immunostaining of dissected posterior midguts from control (*w1118*) flies after DSS treatment/feeding (4% DSS), or mock treatment, or cotreatment/feeding with a Catalase inhibitor (DSS + 3-AT); stained with anti-Mys (green), anti-Dcp-1 (Cleaved Dcp-1; red), and DAPI (blue). Flies were fed (or Mock treated) DSS or DSS + 3-AT for 2 days. (C) Coculture system to study cell death spreading in vitro. UV-induced (100 mJ/cm$^2$) cell death spreading; quantified by percentage of TUNEL-positive cells. S2R+ cells treated with conditioned media (supernatant) containing Mock (control) or Die-V5, and Catalase (3-AT), (D) Images of TUNEL immunostaining from cocultured S2R + cells; UV-induced (100 mJ/cm$^2$) cell death spreading. Cells treated with conditioned media (supernatant) containing Mock (control) or Die-V5, and Catalase inhibitor (3-AT). TUNEL (nuclear, green), DAPI (blue), phalloidin (membrane, red). Data information: Data in panel C are presented as mean ± SD, $n = 9$. $^*P \leq 0.05$ (Student $t$ test). The data underlying the graphs shown in Fig 6C can be found in S1 Data.

the ECM of midgut epithelium (S7B and S7C Fig). DSS-mediated changes in ECM structure correlate with delocalization of the Integrin Mys from the enterocyte–ECM boundary, as well as activation (cleavage) of the effector Caspase Dcp-1 in midgut enterocytes (Fig 6B). These data were confirmed using genetic inhibition (RNAi) of Catalase in enterocytes (NP1G4, TubG80$^{ts}$>UAS-Cat$^{RNAi}$; S7D and S7E Fig).

Finally, in order to explore the synergy between Diedel–Integrin–ECM interactions and membrane-associated Catalase in the regulation of cell death spreading, we returned to our trans-well cell culture system (Fig 3A–3C). Diedel's ability to block the extracellular propagation of UV-induced cell death to non-UV treated "neighbor" cells (and promote survival) is significantly limited when Catalase function is inhibited (using the 3-AT inhibitor; Fig 6C and 6D). These data show that Catalase, and likely extracellular and/or membrane-associated Catalase, is necessary to prevent cell death spreading and promote cell survival driven by Diedel–Integrin–ECM interactions (S7F Fig).

## Discussion

Taken together, these data highlight the synergy between Diedel–Integrin–ECM interactions and membrane-associated Catalase in promoting resilience of epithelial tissues through balancing cellular demise and survival. This synergy prevents the accumulation extracellular ROS (maintaining Redox balance), thus preserving or repairing ECM–Integrin interactions and attenuating the spread of cell death by maintain epithelial cell attachment to the basement membrane. Diedel–Integrin–ECM interactions are therefore critical to limit the burden of adaptive stress responses and mitigate epithelial tissue damage.

Cell death by cell detachment (or anoikis) was originally discovered in vitro using human umbilical vein endothelial cells [79,80]. It was shown that cells grown in suspension undergo cell death that was initiated by a loss of ECM–Integrin interactions. Highlighting the importance of this type of cell death mechanism, resistance to anoikis is likely a vital step during cancer progression within tissues and metastatic colonization [81]. Although this mechanism can be identified in vivo in mammalian epithelial models, it remains difficult to characterize in vivo. The *Drosophila* intestinal (midgut) epithelium provides a reliable genetic model to mechanistically explore epithelial tissue homeostasis. A recent study in *Drosophila* identified the NF-kB–dependent innate immune pathway as a critical regulator of midgut enterocyte shedding (cell detachment) after enteric bacterial infection [82]. Through the identification of secreted Diedel as an ECM protein, present at the basement membrane of epithelial cells, our data highlight a role for ECM–Integrin interactions in the regulation of cell detachment and cell death. We showed that Diedel can bind to *Drosophila* Integrins Mys/If through its RGD domain, and the loss of these interactions can lead to cell detachment and initiation of a Caspase-dependent apoptotic pathway. Although loss of Integrin–ECM interactions leads to the activation of the Caspases, we found that the execution of the cell death occurs only when cells detach from the basement membrane and leave the epithelial layer. This distinct mode of cell death likely aids in the balance of cellular demise and survival, enables tissue resilience, and promotes epithelial tissue integrity (prevents excessive cell death and tissue damage).

Furthermore, using Diedel mutants as a model to mimic the loss of epithelial cell–ECM interaction, we uncovered resilience mechanisms utilized by epithelial tissues to maintain tissue integrity. We found that intracellular Catalase is able to relocalize to the membrane and/or extracellular environment of epithelial enterocytes, likely preventing cell death spreading through elimination of extracellular ROS. In another recent study exploiting *Drosophila* genetics, it was shown that certain midgut enterocytes ("transiently undead" cells that allow for segregation of apoptotic and nonapoptotic functions of Caspases) can recruit Dronc (through

Myo1D) to the NADPH oxidase Duox at the membrane, which drives extracellular ROS production [11]. NADPH oxidases are one of the major generators of cellular/extracellular ROS across taxonomic kingdoms. Additionally, Duox localization at the membrane after Caspase activation can likely lead to cell death spreading through enhancing extracellular ROS [74]. Coupled with our findings presented here, these data suggest that equilibrating extracellular ROS accumulation induced by dying cells (adaptive stress responses) with quenching of extracellular ROS in nearby cells, potentially through membrane-associated Catalase, is critical to promote resilience of epithelial tissues and balance cell death/survival. A similar mechanism was observed in vitro in cancer cells, where translocation of Catalase to the outside of the cell membranes was associated with cellular transformation and tumor progression [68]. Thus, it is likely that resilience mechanisms, such as those driven by membrane-associated Catalase, can be hijacked by cancer cells to resist anoikis and maintain cell–ECM interactions and cell attachment to basement membranes.

Overall, our data shed light on the plasticity of epithelial tissues, where resistance and resilience mechanisms cooperate to maintain tissue integrity and prevent damage. Fundamentally, these mechanisms are crucial when epithelial tissues are presented with internal or external threats (and when cell death is elevated as part of adaptive stress responses) but are also likely to represent mechanisms that partly underlie the etiology of certain diseases (such as cancer).

## Materials and methods

### *Drosophila melanogaster* husbandry and strains

The following strains were obtained from the Bloomington Drosophila Stock Center: *w1118*, Actin-Cas9 (#54590), UAS-Rpr (#5824), Catalase-GFP(#60212), DaGal4 (#55851), GmrGal4 (#1104), UAS-Egr (#51283), UAS-p35 (#5072), UAS-RedStinger (#8545), PplGal4 was a gift from M. Pankratz, NP1Gal4 was a gift from D. Ferrandon, CGGal4 was a gift from C. Thummel, UAS-Hep$^{Act}$, and Sep-Gal4 was a gift from H. Jasper. LanB1-GFP (# v318180) was obtained from Vienna Drosophila Resource Center. UAS-Die$^{RNAi}$ and UAS-Die were previously generated in our lab [15]. Diedel mutant flies (*die$^{Δ1}$* and *die$^{Δ2}$*), Diedel gRNA flies, Die$^P$-DieV5, UAS-Die$^{RGE}$#1, and UAS-DIe$^{RGE}$#2 transgenic flies were generated for this study.

All flies (expect for Diedel mutants) used in this study were backcrossed ×10 into the *w1118* background, with continued backcrossing every 6 to 8 months to maintain isogenecity. Diedel mutant flies (*die$^{Δ1}$* and *die$^{Δ2}$*) were compared to a revertant strain (WT) without underlying changes in gene sequence after CRISPR-Cas9 to match genetic background. All flies were reared on and feed a standard yeast- and cornmeal-based diet at 25˚C and 65% humidity on a 12-hour light/dark cycle, unless otherwise indicated. The diet was made with the following protocol: 14 g agar/ 165.4 g malt extract/ 41.4 g dry yeast/ 78.2 g cornmeal/ 4.7 ml propionic acid/ 3 g methyl 4-hydroxybenzoate/ 1.5 L water. Progeny of crosses were reared in standard fly bottles using this diet. Emerging adult flies were mated and maintained on this diet in standard fly vials (approximately 20 female flies per vial) before dissection or analysis.

For 3-Amino-1,2,4-triazole (3-AT, Acros Organics) feeding/treatment, 3-AT was mixed into the standard lab diet, resulting in a final concentration of 10 μM.

For DSS (dextran sulfate sodium salt colitis grade [MW 36,000 to 50,000]; MP Biomedicals) feeding/treatment, a 4% solution of DSS (dissolved in 5% sucrose) was thatched on top of standard lab food vials. Mock controls were fed just a 5% sucrose solution.

*Female flies were used for all experiments due to sex-specific differences in midgut regeneration.* All adult flies were aged 7 to 10 days before dissection/analysis.

## Generation of transgenic *Drosophila*

*die^{Δ1}* and *die^{Δ2}* mutant *Drosophila* were generated using CRISPR-Cas9. Briefly, a transgenic fly line expressing Cas9 protein using the ubiquitous promoter actin was crossed to a second fly line expressing a custom Diedel guide RNA (gRNA). The cross produces offspring with an active Cas9–gRNA complex, which cleaves and mutates the target site. The following gRNA sequence was used for Diedel mutant flies: GTCAGCCTGCTTCTCGTCGG. To confirm the absence of Diedel protein with underlying base pair changes in *die^{Δ1}* and *die^{Δ2}* mutants, Diedel mutant genomic loci were cloned into the pAC5.1 V5 (using standard Die cloning primers; S2 Table) and transfected/expressed in S2R+ cells. Immunoblotting with anti-V5 was used to confirm the absence of tagged protein in vitro in cell homogenate and the supernatant (secreted).

To generate Die^P-DieV5 transgenic *Drosophila*, we used the plasmids pB-Die^P-RFP and pAC5.1-DieV5 from our previous study [15] and replaced RFP sequence from pB-Die^P-RFP with Diedel cDNA tagged with V5-His from pAC5.1-DieV5. The new plasmid pB-Die^P-DieV5 was injected into *w1118* embryos; attp40 embryos with a phiC31 integrase helper plasmid (Rainbow Transgenic Flies).

To generate UAS-Die^{RGE#1} and UAS-Die^{RGE#2} transgenic *Drosophila*, we used the pAC5.1 Die^{RGE} plasmid generated for this study (see Cf Plasmid Construction section below) and sub-cloned Die^{RGE}-V5 sequence into the pUASt plasmid (DGRC #1000). The new plasmid pUASt-Die^{RGE} was injected into *w1118* embryos (Rainbow Transgenic Flies).

## Conditional expression of UAS-linked transgenes

The NP1Gal4 or PplGal4 drivers were combined with a ubiquitously expressed temperature sensitive Gal80 inhibitor (TubGal80^{ts}). Crosses and flies were kept at 18°C (permissive temperature), and 5-day-old females were shifted to 29°C for 2 days to allow expression of the transgenes and then fed for another 2 days with DSS or sucrose (as describe in *Drosophila melanogaster* husbandry and strains section) at 29°C.

## MARCM analysis

To generate undead cells clones in the midgut, w;UAS-Rpr,UAS-p35;Cat-GFP,FRT82B/TM3 flies were crossed with y,w,hsFLP,UAS-RedStinger3;;tub-gal4, FRT82B,TubGal80/TM6B flies. For the Control clone, w;;Cat-GFP,FRT82B/TM3 flies were crossed with y,w,hsFLP,UAS-RedStinger3;;tub-gal4, FRT82B,TubGal80/TM6B flies. For midgut clone induction, progeny (4 to 5 days post-eclosion and after mating) of the correct genotype were heat shocked at 37°C for 45 minutes. Approximately 5 days after heat shock, midguts were dissected and MARCM clones were analyzed.

## Intestinal permeability ("Smurf") assay

Flies were aged on standard food until the day of the assay. Dyed food was prepared using a standard food recipe supplemented with blue dye no. 1 (Erioglaucine, ER120; Spectrum) at 2.5% (wt/vol). Flies were maintained on dyed food overnight. A fly was counted as a Smurf when the dye coloration could be observed outside the gastrointestinal tract.

## Immunostaining and microscopy in vivo

Intact midguts (flies aged 7 to 10 days) were dissected and fixed at room temperature for 30 minutes in 100 mM glutamic acid, 25 mM KCl, 20 mM MgSO4, 4 mM sodium phosphate, 1 mM MgCl2, and 4% formaldehyde. All subsequent incubations were done in PBS, 0.5% BSA,

and 0.1% Triton X-100 at 4°C. The following primary antibodies were used: mouse anti phospho-Histone 3 (Cell Signaling 9701, 1:1,000), cleaved Cas-3 (Cell Signaling 9661, 1:100), cleaved Dcp-1 (Cell Signaling 9578, 1:1,000), Phospho-ERK (Cell Signaling #4370; 1:1,000), His-tag (Cell Signaling #12698 1:1,000) Catalase H9 (Santa Cruz SC-271803, 1:100), and V5 tag (Sigma V8137, 1:500), Flag Tag M2 (F1804 Sigma, 1:500), anti-GFP 3E6 (Thermo Fisher # A-11120 1:100). Mouse anti-Mys CF.6G11 (1:100), Mouse Anti-Nub 2D4 (Pdm1) (1:100), and mouse anti-Dlg1 4F3 (1:100) were acquired form the Developmental Studies Hybridoma Bank. Fluorescent secondary antibodies were obtained from Jackson Immunoresearch. Hoechst (1:1,000) was used to stain DNA.

For Talin immunostaining, intact midgut were dissected in PBS and fixed with 4% paraformaldehyde for 20 minutes at room temperature, washed 3 times with PBS containing 0.1% Triton X-100 (PBST), then midguts were incubated with boiled citrate buffer (10 mM sodium citrate, 0.05% Tween 20 (pH 6.0)) at room temperature for 15 minutes, washed 3 times with PBS then blocked in blocking buffer (5% BSA in PBST) for 1 hour. Primary antibody mouse anti-Talin A22A (1:100, acquired from Developmental Studies Hybridoma Bank) was applied overnight at 4°C. Alexa Fluor–conjugated secondary (Jackson Immunoresearch, 1:500) antibodies were incubated for 2 hours at room temperature. Hoechst was used to stain DNA, and Phalloidin (Alexa 633 or Rhodamin from Thermo Fisher [A22284, R415]) was used to stain F-Actin fibers.

For extracellular thiol cysteine staining (cysteine oxidation), intact midguts were dissected in PBS+Dimedone (10 mM Dimedone), then incubated for 30 minutes at room temperature in PBS+Dimedone, washed 2 times with PBS then fixed with 4% paraformaldehyde for 20 minutes at room temperature, and then blocked with PBS+0.5% BSA for 1 hour. Primary cysteine (sulfonate) polyclonal antibody (ADI-OSA-820-D, Enzo, 1:500) was incubated overnight in PBS, 0.5% BSA at 4°C. Alexa Fluor–conjugated secondary (Jackson Immunoresearch, 1:500) antibody was incubated for 2 hours at room temperature. Hoechst was used to stain DNA, and Phalloidin (Alexa 633 from Thermo Fisher [A22284]) was used to stain F-Actin fibers.

For midguts stained in nonpermeable conditions, Triton was omitted from buffers until the end of incubation with secondary antibodies. Triton was reintroduced to stain with Hoechst and Phalloidin.

Confocal images were collected using a Nikon Eclipse Ti confocal system (utilizing a single focal plane) and processed using the Nikon software.

## Developmental timing

Around 75 virgin flies were mated with 50 males from mutant lines, balanced with TM3, Act-GFP, and were allowed to lay eggs for less than 24 hours on apple agar (apple juice + agar) plates, supplemented with live yeast paste. After 24 hours, 50 L1 homozygote and heterozygote larva were sorted by genotype based on the expression of GFP and then transferred to apple agar plates with yeast. The number of pupa and adult flies was counted.

## Cell culture, transfection, and coculture

*Drosophila* S2 cells R+ (S2R+, acquired from the Drosophila Genomics Resource Center) were maintained in Shields and Sang M3 + BPYE media supplemented with 10% FCS, 50 U/ml penicillin, and 50 μg/ml streptomycin at 25°C. To induce apoptosis/RCD, cell media was removed, and cells ($2 \times 10^6$ cell/mL in petri dish) were treated with ultraviolet C light (UVC; dose as indicated and previously described in [15]). For immunostaining, cells were plated on coverglass coated with poly-L-lysine and treated as describe below.

Coculture experiments were performed using 12-well plates with trans-well insert with a 3-μm pore size (VWR, 29442–080). For 3-AT (Acros Organics) treatment, 3-AT was added to the culture media, resulting in a final concentration of 10 μM.

## Primary cell culture

Embryonic primary cell cultures were isolated from gastrulating Catalase-GFP (BDSC#60212) embryos as described previously [83]. Briefly, embryos were collected on apple agar (apple juice + agar) plates with yeast paste and incubated for another 3.5 hours at 25°C. Embryos were washed extensively with H2O, then dechorionized with bleach (3% sodium hypochlorite) for 3 minutes, and then washed with water to remove residual bleach and again washed with 75% ethanol. Embryos were then homogenized with a Dounce homogenizer in Schneider's medium supplemented with 10% FBS, 50 U/ml penicillin, and 50 μg/ml streptomycin. Dissociated cells were filtrated through a 40-μm cell strainer, and then centrifuged at 2,000 rpm for 5 minutes. Cells were washed and centrifuge again, and then resuspended in medium. For immunostaining, cells were plated on coverglass coated with poly-L-lysine and treated as describe below.

## In vitro immunostaining

Cells were plated on coverglass coated with poly-L-lysine overnight before treatment. For immunostaining, the cells were washed with PBS and then fixed with 4% formaldehyde for 20 minutes, followed by 2X PBS washes. All subsequent incubations were performed in PBS, 0.5% BSA, and 0.1% Triton X-100 at 4°C. The following primary antibodies were used; anti-V5 (Sigma V8137, 1:500) and anti-Catalase H9 (Santa Cruz SC-271803, 1:100). Fluorescent secondary antibodies were obtained from Jackson Immunoresearch. Phalloidin was used to stain membranes and Hoechst (1:1,000) was used to stain DNA.

## Proteomic analysis

S2R+ cells ($2 \times 10^6$ cell/mL in petri dish) were treated with UV (100 mJ/cm$^2$) and subsequently cultured in the presence of conditioned media containing Die-V5-His (or Mock) for 5 hours. Media was removed and cells washed with cold PBS, then crosslinked with 4% paraformaldehyde for 5 minutes on ice, washed X3 with cold PBS, and then lysed with RIPA buffer. Proteins were pulled down using Pierce HisPur Ni-NTA Spin Columns. The elution fractions were electrophoresed in 12% SDS-PAGE, and protein bands were stained with Coomassie blue. Bands were excised and subjected to in-gel trypsin digestion followed by identification.

All MS/MS samples were analyzed using Mascot (Matrix Science, London, UK; version 2.7.0) and X! Tandem (The GPM, thegpm.org; version CYCLONE (2010.12.01.1)). Mascot was set up to search the NCBIprot_20170227 database, assuming digestion with the enzyme trypsin. Mascot and X! Tandem were searched with a fragment ion mass tolerance of 0.80 Da and a parent ion tolerance of 20 PPM. Alternatively, the total elution fraction was subjected the same protocol.

## Criteria for protein identification

Scaffold (version Scaffold_4.9.0, Proteome Software, Portland, Oregon) was used to validate MS/MS-based peptide and protein identifications. Peptide identifications were accepted if they could be established at greater than 95.0% probability by the Scaffold Local FDR algorithm. Protein identifications were accepted if they could be established at greater than 99.0% probability and contained at least 2 identified peptides. Protein probabilities were assigned by

the Protein Prophet algorithm [84]. Proteins that contained similar peptides and could not be differentiated based on MS/MS analysis alone were grouped to satisfy the principles of parsimony.

## Plasmid construction

pAC5.1 Die^RGE-V5 plasmid was generated from the pAC5.1 Die-V5 using overlapping PCR. Briefly, 2 fragments were generated by PCR using pAC5.1 Die-V5 plasmid as template with the following primers: Fragment1 F1: taaggtaccatggcatccccagtagtc, R1: ctcgca**c**tcgcctctgtccatc; Fragment2 F2: gatggacagaggcga**g**tgcgag, R2: taactcgagaaatggcagcctggt; introducing a mutation in the RGD domain of Diedel, the fragments were then gel purified and fused together using the overlap extension polymerase chain reaction with the Phusion enzyme (NEB, Ipswich, Massachusetts , USA). The fused fragment with the mutation in the RGD motif was isolated with electrophoresis and gel purified, then cloned into the pAC5.1 V5-His plasmid using the restriction enzymes XhoI and KpnI. Integrins Inflated (If) and Myospheroid (Mys) were amplified from cDNA of *w1118* L3 larva using specific primers (see S2 Table) and cloned into pMT-puro (Addgene plasmid; #17923).

pAC5.A Flag-Cat plasmid was generated by PCR amplification using specific primer with a Flag tag sequence in 5′ (see S2 Table). Chimeric receptors CD8a and Mys were generated by overlapping PCR (as describe below) using genomic DNA of flies expressing CD8a-mCherry (Bloomington #27391) and the Pmt-Myospheroid plasmid using specific primers (see S2 Table) and then cloned into pMT-Puro plasmid.

## Integrin binding ("cell spreading") assay

S2R+ cells were transfected with pmt-puro-If and selected with puromycin (2 μg/mL), and expression of Inflated (If) Integrin was induced with 2 mM CuSO4 for 24 hours. Cells were then collected and washed with PBS and treated with Trypsin for 30 minutes at room temperature to remove extracellular proteins; the cells were washed with PBS and were allowed to "spread" for 6 hours on coated coverglass with Mock, DieV5, or DieRGE-V5–treated conditioned media. Cells were then fixed for 30 minutes in 4% formaldehyde, permeabilized with PBS+Triton, and stained with Phalloidin and Hoescht. Experiments were done in triplicate with around 100 cells counted in each experiment.

## In vitro dsRNA treatment

Regions of cDNA for Catalase, Diap-1, and RFP (control) were amplified by PCR from plasmids containing cDNA sequences. Each primer used in the PCR contained a T7 polymerase binding site (GAATTAATACGACTCACTATAGGG AGA) at the 5′ and 3′, followed by sequences specific for: Catalase (forward: ATCGCCTTCAGTCCCGCTCA, reverse: GGCCGAAGTTGTCCTCGGTGT), Diap-1 (GACGCTGGAGATGAGGGAGC, reverse: CTGGTGGGGCTTCTGTTTCAGGT), and RFP (forward: ATAGTCTTCTTCTGCATTAC, reverse: AGGACGTCATCAAGGAGTTC). The PCR products were gel purified, and complementary RNA strands were synthesized by in vitro transcription using the T7 Megascript Kit (Ambion Austin, Texas, USA). To obtain dsRNA, sense and antisense strands were annealed by heating to 65°C for 15 minutes and then allowed to cool to room temperature. The quality of the dsRNA was analyzed on an agarose gel and stored at −80°C. For knock-down experiments, S2R+ cells were seeded in 6-well plates in 2 ml of medium the day before the RNAi procedure. The following day, dsRNA (10 μg) transfected using Effectene Transfection Reagent (Qiagen, Hilden, Germany). The S2R+ cells were incubated for 2 days to induce posttranscriptional gene silencing.

### *Drosophila* midgut decellularization and western blotting

Detergent-enzymatic treatment to decellularize intestine was previously established for rat small bowel [85,86] and was optimized for *Drosophila* in this study. Around 10 midguts were dissected in PBS and collected in Eppendorf tube. The midguts were decellularized with 4% sodium deoxycholate for 30 minutes at room temperature, and this was followed by 4 washes with Milli-Q water using a table centrifuge to pellet the midguts. The midguts were then incubated with Pierce Universal Nuclease (250 U/mL) in 1 M NaCl for 30 minutes at room temperature. The decellularized midguts were washed 2 times with Milli-Q Water and resuspend in 20 μL of Laemmli buffer with 1/10 2-mercaptoethanol and boiled for 10 minutes. Proteins were loaded onto 10% stain-free SDS polyacrylamide gels (Bio-Rad) that allow visualization of protein in gel, and then transferred onto nitrocellulose membranes and blocked for 1 hour in TBST (50 mM Tris-HCl (pH 7.5), 150 mM NaCl, 0.05% Tween 20) containing 5% BSA. Protein immunoblots were performed using antibodies against γ-Laminin (D-3) (Santa Cruz, sc-17751). The immobilized proteins were further incubated with StarBrightBlue 520 mouse secondary antibody (Bio-Rad Hercules, California, USA), and protein bands were visualized using a Chemidoc MP Imaging System (Bio-Rad).

### In vitro cell death measurement

For TUNEL assay, S2R+ cells were fixed on a poly-L-lysine (VWR)-coated glass coverslip with 4% formaldehyde for 10 minutes and washed 2 times with PBS. Cell death was detected by using an In Situ Cell Death Detection Kit (Roche, Basel, Switzerland) according to manufacturer's protocol.

For Annexin V staining, S2R+ cells were washed twice with binding buffer (10 mM HEPES, 140 mM NaCl, 2.5 mM CaCl2 (pH 7.4)) and then incubated for 15 minutes in the dark with Annexin V FITC Conjugated (Thermo Fisher, #A13199), according to manufacturer's protocol. After 2 washes with binding buffer, the cells were fixed with 4% PFA.

### In vivo TUNEL (cell death) assay

Intact midguts were dissected (at indicated ages) in 1X PBS and fixed in 4% paraformaldehyde for 25 minutes at room temperature. Samples were washed with 1X PBT for 30 minutes. Cell death was detected by using an In Situ Cell Death Detection Kit (Roche) according to manufacturer's instruction.

### Pupal UV irradiation and phenotype quantification

Mid-aged pupae (24 hours after puparium formation) were collected and subjected to surgical removal of the pupal shell surrounding the head area. UV irradiation was carried out on larvae that were immobilized on the side, so that only one retina was exposed to UV. A UV crosslinker (Stratalinker, 1800) was used with energy set at 20 mJ/cm$^2$. After irradiation, pupae were kept in the dark. The images of UV-damaged adult eyes were taken, and the boundary of each eye was outlined using Photoshop. The eye size was determined by measuring the number of pixels contained within this area. Ratios between the area of irradiated and non-irradiated eyes were then determined.

### Immunoprecipitation assay

S2R+ cells were washed with ice-cold PBS and lysed for 20 minutes on ice in lysis buffer (0.1% SDS, 150 mM NaCl, 1% Triton, 10 mM HEPES, 2 mM EDTA, 50 mM NaF, 2 mM Na3VO4, 0.1% sodium deoxycholate, 1.33 mM Na2HPO4 (pH 7.4) with 10 mM NaH$_2$PO$_4$) supplemented with antiprotease. Lysates were precleared and then incubated overnight at 4°C with

the indicated antibody and Recombinant Protein G—Sepharose 4B (Thermo Fisher #101241). Immunoprecipitates were washed twice with lysis buffer and then twice with Tris buffer (pH 7.5) containing 500 mM NaCl, 1% triton, followed by 2 washes with 10 mM Tris (pH 7.5) containing 250 mM NaCl. Proteins were eluted from beads with SDS sample buffer and boiled for 10 minutes, then separated by SDS polyacrylamide gel electrophoresis, transferred to nitrocellulose membrane, and blotted with the indicated antibody.

## Supporting information

**S1 Fig. Diedel mutant analysis and tissue-specific Diedel production.** (A) Genomic DNA sequence alignment of wild-type (WT) Diedel and CRISPR-Cas9–induced Diedel mutants (*dieΔ1* and *dieΔ2*). White box highlights gRNA sequence; nucleotides represented in blue highlight a mismatch; dashed lines highlight nucleotide deletions. (B) Western blot with anti-V5 antibody of cell lysate or conditioned media (supernatant) from S2R+ cells either nontransfected (Mock) or transfected with a plasmid containing Die-V5, or DieΔ1-V5, or DieΔ2-V5. Die-V5 band is highlighted by black arrow. (C) Histogram representing percentage of L1 larva (from different Diedel mutant genotypes) reaching pupariation. (D) Histogram representing percentage of L1 larva (from different Diedel mutant genotypes) reaching adult stage (eclosion). (E) Pdm1 immunostaining of dissected posterior midgut from control (WT, revertant [rev.]), Diedel mutant homozygote (*dieΔ1/dieΔ1*), and Diedel mutant trans-heterozygote (*dieΔ1/dieΔ2*) flies; stained with anti-Pdm1 (green), Phalloidin (Phall.; purple), and DAPI (blue). (F) Die-V5 levels in dissected *Drosophila* fat bodies from controls (negative control; *w1118*) and *w1118*; Die$^P$-Die-V5 / Die$^P$-Die-V5 transgenic flies; stained with anti-V5 antibody (Die-V5; red) and DAPI (blue). (G) Die-V5 and Laminin B1 (LanB1) immunostaining of dissected posterior midguts from endogenously tagged Laminin B1 (LanB1-GFP) transgenic flies and LanB1-GFP/Die$^P$-Die-V5 flies; bottom panel represents cross-section (apical–basal polarity is highlighted with arrow); stained with anti-His (Red), anti-GFP (purple), and DAPI (blue). The data underlying the graphs shown in S1C and S1D Fig can be found in S1 Data. (TIF)

**S2 Fig. Diedel-dependent control of Integrin Mys localization.** (A) Integrin Mys immunostaining of dissected posterior midgut from control (WT, revertant [rev.]) flies; panels represent cross-sections (apical–basal polarity is highlighted with arrow); stained with anti-Mys (green), Phalloidin (Phall.; purple), and/or DAPI (blue). (B) Integrin Mys immunostaining of dissected posterior midgut from control flies (*w1118*; PplGal4,TubG80$^{ts}$) and flies with conditional (adult-specific) fat body attenuation of Diedel (*w1118*; PplGal4,TubG80$^{ts}$ / UAS-Die RNAi) after 7 days at 29˚C; cross-sections (apical–basal polarity is highlighted with arrow); stained with anti-Mys (green), Phalloidin (Phall.; purple), and/or DAPI (blue). (C) Integrin Mys immunostaining of dissected posterior midguts from control flies (*w1118*; CGGal4) and flies with fat body expression of Diedel RGE (*w1118*; CGGal4 / UAS-Die$^{RGE\#2}$: transgenic line 2), stained with anti-Mys (green), Phalloidin (Phall.; red), and/or DAPI (blue). (TIF)

**S3 Fig. Diedel in the regulation of cell death and intestinal barrier function.** (A) Schematic of the intrinsic and extrinsic (TNF) apoptosis-induced RCD pathway in *Drosophila*. (B) UV induced retinal apoptosis. Photomicrographs of adult eyes from control flies (*w1118*) or flies ubiquitously overexpressing Diedel (w1118; DaGal4 / UAS-Die) During pupal development, retina were either UV irradiated (20 mJ/cm$^2$, right eye; black arrow) or not (left eye). (C) Histogram representing the ratio between the eye area of the irradiated eye (right, R) to the non-irradiated (untreated) eye (left, L) of the genotypes described above. Bars represent mean ± SE,

*n* = 7. (D) Constitutive activation of JNKK (UAS-Hep^ACT) in the developing fly retina (Sep-Gal4; expressed in photoreceptors and cone cells) results in a "rough" eye phenotype induced by apoptosis. Overexpressing Diedel (*w1118*; SepGal4, UAS-HepACT / UAS-Die) does not change this phenotype compared to controls (*w1118*; SepGal4, UAS-HepACT). (E) Overexpression of Reaper (UAS-Rpr) in the developing fly retina (GMRGal4) results in an apoptosis-induced cell death. Overexpressing Diedel (*w1118*; GMRGal4, UAS-Rpr / UAS-Die) does not change this phenotype compared to controls (*w1118*; SepGal4, UAS-Rpr). (F) Overexpression of Eiger (UAS-Eiger) in the developing fly retina (GMRGal4) results in an apoptosis-induced cell death rescued by Eiger inhibition (GMRGal4, UAS-Eiger / UAS-Eiger RNAi). Overexpressing Diedel (*w1118*; GMRGal4, UAS-Eiger / UAS-Die) does not change this phenotype compared to controls (*w1118*; GMRGal4: UAS-Eiger). (G) Intestinal barrier dysfunction ("Smurf" assay) in control (WT, revertant [rev.]), Diedel mutant homozygote (*dieΔ1/dieΔ1*) during aging. ND, nondetected, *n* = 100 flies/group. (H) Quantification of ISC proliferation (mitoses per whole dissected midgut); assayed by anti-pH3 immunostaining in control (WT, revertant [rev.]), Diedel mutant homozygote (*dieΔ1/dieΔ1*) and Diedel mutant trans-heterozygote (*dieΔ1/dieΔ2*) flies. Bars represent mean ± SE, *n* = 10–15. *$P \leq 0.05$ (Student *t* test). (I) Representative images of TUNEL immunostaining (posterior midgut) at indicated genotypes; TUNEL (nuclear, green), DAPI (blue). The data underlying the graphs shown in S3G and S3H Fig can be found in S1 Data. ISC, intestinal stem cell; RCD, regulated cell death; TNF, tumor necrosis factor; WT, wild type.
(TIF)

**S4 Fig. Caspase localization in the midgut epithelium.** (A) Integrin Mys and Caspase (cleaved Dcp-1) immunostaining of dissected posterior midgut of control (WT, revertant [rev.]) and Diedel mutant (*dieΔ1/dieΔ1*) flies; stained with anti-Mys (green), anti-Dcp-1 (Cleaved Dcp-1; red), and DAPI (blue). Images represent 3D Confocal Z-Stack representation of Integrin Mys and Cleaved Dcp-1 localization. Apical–basal polarity/arrangement in sections is highlighted. (B) Disc large 1 (Dlg1; septate junction marker) and Caspase (cleaved Dcp-1) immunostaining of dissected posterior midgut of control (WT, revertant [rev.]) and Diedel mutant (*dieΔ1/dieΔ1*) flies; stained with anti-Dlg (green), anti-Dcp-1 (Cleaved Dcp-1; red), and DAPI (blue).
(TIF)

**S5 Fig. Diedel promotes cell survival in the midgut epithelium.** (A) Schematic of CD8-alpha and Integrin Mys chimeric receptors. pMT CCM expresses the extracellular and transmembrane region of the mouse CD8-alpha and the intracellular region of Integrin Mys; pMT CMM expresses the transmembrane and intracellular region of Integrin Mys and the extracellular region of CD8-alpha (B-C) Cleaved DCP1 and CD8-alpha immunostaining of S2R+ cells transfected with chimeric receptors (B) pMT CCM and (C) pMT CMM. Chimeric receptor (plasmid) expression was induced by CuS04 at 3 concentrations, 2 mM, 4 mM, and 6 mM, and stained with anti-cleaved DCP1 (green), anti-CD8-alpha (Cleaved Dcp-1; red), and DAPI (blue). (D) phospho-ERK immunostaining of dissected posterior midgut from control flies (*w1118*; PplGal4,TubG80^ts), flies with conditional (adult-specific) fat body attenuation of Diedel (*w1118*; PplGal4,TubG80^ts / UAS-Die RNAi), or conditional (adult-specific) fat body overexpression of Diedel (*w1118*; PplGal4,TubG80^ts / UAS-Die) after 5 days at 29°C; stained with anti-phoso-ERK (green), anti-Arm (armadillo; red), and DAPI (blue).
(TIF)

**S6 Fig. Catalase associates with cellular membranes/ECM after stress.** (A) Catalase and anti-V5 immunostaining of S2R+ cells after UV exposure (100 mJ/cm^2); cells treated with Die-V5 conditioned media (supernatant) or Mock (control); stained with anti-V5 antibody (Die-

V5; red) and anti-Catalase (green). Cells were with detergent in order to visualize intracellular proteins. (B) Schematic representation of the EFGP-multiTAG insertion into the *Drosophila* Catalase gene loci. **(C)** Co-immunoprecipitation of Die-V5 and Cat-FLAG in S2R+ cells transfected with pAC.5.1 Cat-Flag or cotransfected with pAC.5.1 Cat-Flag and pAC5.1 Die-V5; +/− UV: 100 mJ/cm$^2$. Western blot using anti-Flag antibody and anti-V5 antibody. Bottom panels display entire images of western blots. (D) Membrane/ECM cysteine oxidation of dissected posterior midguts of *w1118* (Ctrl.) flies; DSS treatment/feeding (4% DSS), or mock treatment, or cotreatment/feeding with a Catalase inhibitor (DSS + 3-AT); stained with anti-Cysteine (oxidized, Cys. Ox; red) and DAPI (blue). Immunostaining was performed in non-permeabilized conditions to visualize membrane/ECM cysteine oxidation. Flies were fed (or Mock treated) DSS or DSS + 3-AT for 2 days.
(TIF)

**S7 Fig. Membrane-associated Catalase is necessary for ECM preservation / repair after epithelial tissue damage.** (A) Brightfield microscopy images of *w1118* (WT; control) *Drosophila* posterior midguts before and after decellularization. (B) Western blot with anti-Laminin B2 antibody (LanB2, top panel) and ECM protein composition shown by stain free gel (bottom panel); after midgut decellularization. *w1118* (WT; control) midguts were dissected after DSS treatment/feeding (4% DSS, +), or mock treatment, or cotreatment/feeding with a Catalase inhibitor (DSS + 3-AT). Flies were fed (or Mock treated) DSS or DSS + 3-AT for 2 days. (C) Entire image of western blot; related to S7B Fig top panel. Western blot with anti-Laminin B2 antibody after midgut decellularization.(D) Catalase immunostaining of dissected posterior midgut of control flies (*w1118*; NP1Gal4, tubG80ts) or flies with adult enterocyte-specific inhibition of Catalase (*w1118*; NP1Gal4, tubG80ts / UAS-Catalase [Cat] RNAi); stained with anti-Catalase (green) and DAPI (blue). Immunostaining confirms RNAi efficacy. (E) Integrin Mys immunostaining of dissected posterior midgut of control flies (*w1118*; NP1Gal4, tubG80ts) or flies with adult enterocyte-specific inhibition of Catalase (*w1118*; NP1Gal4, tubG80ts / UAS-Catalase [Cat] RNAi); midguts were dissected after DSS treatment/feeding (+ 4% DSS) and stained with anti-Mys (green) and Phalloidin (Phall.; red). Panels represent cross-sections (apical–basal polarity is highlighted with arrow). (F) Proposed model depicting the role of Diedel–Integrin–ECM interactions and membrane-associated Catalase in the regulation of extracellular ROS to promote epithelium resilience.
(TIF)

**S1 Table. Mass spectrometry protein identification after pulldown with Mock or Diedel-V5.**
(XLSX)

**S2 Table. Primers.**
(XLSX)

**S1 Data. Raw data.**
(XLSX)

**S1 Raw Images. Raw images.**
(PDF)

## Acknowledgments

We would like to thank Dr. Larry Dangott (Protein Chemistry Lab at Texas A&M University) for help with mass spectrometry and proteomics.

## Author Contributions

**Conceptualization:** Mohamed Mlih, Jason Karpac.

**Data curation:** Mohamed Mlih.

**Formal analysis:** Mohamed Mlih.

**Funding acquisition:** Jason Karpac.

**Investigation:** Mohamed Mlih, Jason Karpac.

**Methodology:** Mohamed Mlih, Jason Karpac.

**Resources:** Jason Karpac.

**Supervision:** Jason Karpac.

**Validation:** Mohamed Mlih, Jason Karpac.

**Visualization:** Mohamed Mlih, Jason Karpac.

**Writing – original draft:** Mohamed Mlih, Jason Karpac.

**Writing – review & editing:** Mohamed Mlih, Jason Karpac.

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
