## [Editor Report · Decision Letter 0]

2 Sep 2021

Dear Dr Karpac, 

Thank you for submitting your manuscript entitled "Membrane-associated Catalase protects Integrin-ECM interactions and promotes resilience of the Drosophila intestinal epithelium" for consideration as a Update Article by PLOS Biology.

Your manuscript has now been evaluated by the PLOS Biology editorial staff as well as by an academic editor with relevant expertise and I am writing to let you know that we would like to send your submission out for external peer review.

Please re-submit your manuscript within two working days, i.e. by Sep 06 2021 11:59PM.

Kind regards,

Ines

--

Ines Alvarez-Garcia, PhD

Senior Editor

PLOS Biology

---

## [Decision Letter · Decision Letter 1]

4 Oct 2021

Dear Dr Karpac,

Thank you for submitting your manuscript entitled "Membrane-associated Catalase protects Integrin-ECM interactions and promotes resilience of the Drosophila intestinal epithelium" for consideration as a Update Article at PLOS Biology. Your manuscript has been evaluated by the PLOS Biology editors, an Academic Editor with relevant expertise, and by three independent reviewers.

You will see that the reviewers find your conclusions novel and interesting, however they also think that the experiments involving the interaction of Diedel with Catalase and how oxidative damage of ECM is regulated need to be strengthen in order for us to pursue the manuscript further for publication. They all propose several experiments to address these issues and other concerns that should be performed.

In light of the reviews (attached below), we will not be able to accept the current version of the manuscript, but we would welcome re-submission of a revised version that takes into account the reviewers' comments. We cannot make any decision about publication until we have seen the revised manuscript and your response to the reviewers' comments. Your revised manuscript is also likely to be sent for further evaluation by the reviewers.

We expect to receive your revised manuscript within 3 months. 

**IMPORTANT - SUBMITTING YOUR REVISION**

3. Resubmission Checklist

a) *Published Peer Review*

b) *PLOS Data Policy*

d) *Blurb*

Please also provide a blurb which (if accepted) will be included in our weekly and monthly Electronic Table of Contents, sent out to readers of PLOS Biology, and may be used to promote your article in social media. The blurb should be about 30-40 words long and is subject to editorial changes. It should, without exaggeration, entice people to read your manuscript. It should not be redundant with the title and should not contain acronyms or abbreviations. For examples, view our author guidelines: https://journals.plos.org/plosbiology/s/revising-your-manuscript#loc-blurb

Sincerely,

Ines

--

Ines Alvarez-Garcia, PhD

Senior Editor

PLOS Biology

Reviewers’ comments

Rev. 1:

This study identifies a novel mechanism of the Drosophila gut tissue against apoptotic cell death. It revealed how Diedel secreted from the fat body protects enterocytes via its binding ability to Integrin through the RGD domain. Through the interactome analysis, the authors further identified that Catalase is bound at the membrane and preserve the gut tissue. This relocalisation of Catalase inhibits the spreading of the apoptosis by reducing ROS as an adaptive stress response. The study has been done well and the manuscript are carefully written honestly. Overall, the data provided support the conclusion, and their finding is interesting.

Major comments,

1, Throughout the manuscript, the authors use either whole body mutant (It is nice that the authors characterise the phenotype well) or RNAi, by which they cannot deny the possible effect on the development of the gut. They need to do some adult-specific manipulation, which would tell more how adult homeostasis is perturbed by the temporal loss of Diedel/Catalase function in the adult gut. The time course analysis would further strengthen the description of the phenotype.

2, It is not clear for this reviewer how Diedel, Integrin, or catalase localisation is regulated.

2-1, Fig.1F suggests the binding of Die to gut (in normal condition), but for S2 cells, the Die binding is UV-dependent (Fig. 3A, 4D). How has UV-stimulation enabled Die to be at the membrane?

2-2, Why Integrin is not identified in the interactor of Die-V5? Related to 2-1, Is it because S2 cells do not express enough integrins? Please discuss this point.

2-3, In Fig. 5, the authors claimed that Catalase is localised to the membrane upon DSS feeding, but it is interesting whether Die is required for this phenotype. Whether Die mutant fails to show this Catalase relocalisation is vital step to draw the conclusion.

3, One of the most attractive points of the manuscript would lie in the "cell death spreading" protected by Die. In Fig. 3B-D, together with Fig. S2, the authors tried to show the non-cell-autonomous killing of the cell is affected by Die. However, it is not directly tested whether this is the case in the gut tissue. In S2 cells, the type of cell death induced here is not well understood (is it apoptosis? through what kind of secreted protein?). It is technically not difficult to test the contribution of cell death-inducing cell death by overexpressing rpr/hid in the gut in a clonal manner. Also, they can probably use the model of apoptosis-induced apoptosis in the wing discs (for example, Perez-Garijo et al., elife, 2013).

4, Did the authors test the expression of Die in the gut? In Fig. 1F and S1E, they use DieP-DieV5, but they might use fat body Gal4 to express Die V5 to see whether Die is from the fat body. Also, they can use Die-/- plus FB>Die (and the negative control DieRGD) to beautifully show how fat body-derived Die is a key for their phenotype through its binding to Integrin.

Rev. 2:

This is a review of the manuscript "Membrane-associated Catalase protects Integrin-ECM interactions and promotes resilience of the Drosophila intestinal epithelium" by Mlih and Karpac. In this work, the authors begin by exploring the function of the secreted signal Diedel and end by looking at a potential stress-induced change in catalase localization.

The first part of the paper, on Diedel function, is generally good, and appears well-done. They analyze a Diedel mutant they generated, and then go on to show that Diedel appears to function, at least in part, by binding integrins. I have no significant concerns about this section.

The second part of the paper focuses on catalase localization and oxidative damage to epithelia, with which Diedel is supposed to interact. This part of the paper contains several interesting observations, but these appear to be held together by a series of untested assumptions. A large amount of experimental work would be required to bring the second section up to a publishable standard.

Major issues:

1. One of the major points of the second part of the paper is that the authors observe what appears to be extracellular catalase. The implicit assumption is that catalase is somehow being secreted by stressed epithelial cells. However, it's not clear that this is how this system would have to work: for example, catalase could end up in the extracellular space as a result of cell lysis. The microscopy shows what appears to be a vesicular localization of catalase under stress; this could be catalase being endocytosed or secreted.

2. More proof is required that what we are observing in these figures really is extracellular catalase. All we now have is immunostaining on non-permeabilized tissue.

3. The IP for Diedel-interacting proteins in Figure 4B/C and Table S1 is interesting, but there are clearly a very large number of false positives in this analysis. Without validation (for example, conventional co-IP followed by Western blot), this assay doesn't really support much inferential weight. The histogram in 4C also appears to be mislabeled, making its interpretation difficult.

4. It's an important part of the authors' story that Diedel and catalase have a meaningful interaction after tissue damage. However, they don't assay whether the interaction they observe in Figure 4A depends on irradiation or not. This should be shown.

5. Ultimately, the authors want to believe that the relocalization of catalase they observe from the intracellular to the extracellular space is functionally required for control of oxidative damage of ECM; this is explicitly stated in the abstract ("Intracellular Catalase can re-localize to the extracellular membrane and limit ECM damage induced by the amplification of extracellular ROS, a critical adaptive stress response"). However, the requirement for extracellular catalase is not demonstrated.

Minor point:

The majority of the figures are immunofluorescent images rendered in color-blind unfriendly palettes. Red-green colorblind readers cannot distinguish green from yellow or blue from purple.

Rev. 3:

In this manuscript, the authors characterize a secreted protein, Diedel (die), which they have identified in previous work, and link its activity to resilience of epithelial cells, mostly at the adult midgut. Diedel is secreted from fat body and inhibits apoptosis-induced regulated cell death (RCD). Here, they generate two mutants of die and find that the integrity of the epithelial architecture and integrity of the midgut is disrupted. By using a tagged transgene they localize Diedel to the basement membrane of the midgut. The authors then provide evidence that Diedel interacts with the integrin Mys via its RGD motif and is required for attaching epithelial cells to the extracellular matrix. The authors then move on to examine how Diedel links the integrity of the epithelium to its control of apoptosis-induced RCD. In a trans-well cell culture system they show that Diedel inhibits apoptosis-induced RCD through maintaining the ECM-integrin interaction. Finally, the authors identify Catalase as an interacting protein of Diedel in UV-treated S2 cells. What follows is - in my mind - a rather confusing attempt to link redox balance mediated by Catalase with tissue resilience after stress.

Overall, this manuscript has strengths and weaknesses. A major strength is the finding that Diedel can act as a ligand for the integrin Mys. However, other sections of the manuscript are rather weak, vague and often confusing.

1. In figure 1, the authors claim that Diedel localizes to the basement membrane. However, they never use any marker of the basement membrane to directly demonstrate this localization. They also claim that in die mutants, enterocytes detach from the basement membrane. Again, without proper markers, these conclusions are hard to justify.

2. The description of the cleaved Dcp-1 staining in figure 3 was not well presented. First, the authors seem to believe that caspases are nuclear (see lines 319 and 328). Caspases can be nuclear, but they are largely cytosolic. So, the cDcp1 staining should by mostly cytosolic. The membrane localization of Dcp-1 is interesting and there is precedence for that, but to consider the possibility that a cytosolic protein can interact with the extracellular basement membrane (line 321) is rather strange. Second, although the authors show increased caspase activity in die mutants, they do not show whether there is more apoptosis. How does TUNEL labeling or other apoptosis assays look like in die mutants?

3. The co-localization data of Dcp1 with Dlg1 in figure S3E was not present in my copy of the manuscript.

4. The section about the characterization of Catalase is very confusing. The authors state that they identified cytosolic Catalase as an interacting protein of extracellular Diedel. I understand that similar to Dcp1 above, a cytosolic protein can be membrane localized under certain circumstances. However, how can extracellular Diedel interact with intracellular protein. In lines 391/2, they state that Catalase is extra-cellular and the non-permeabilized conditions of the experiments further supports the extracellular location. At another point in the manuscript (lines 415/6), the authors state that Catalase acts on extracellular ROS. How can that be? Can Catalase cross membranes? I also find the conclusion of this section that "these molecular changes underly resilience mechanisms in epithelial tissues" (lines 422/3) very vague and over-interpreted.

5. The trans-well cell culture experiments are quite interesting. How is apoptosis transmitted from UV-treated cells to untreated cells? Is there a death signal? The authors must be aware of the work by Perez-Garijo et al. (2013) in eLife which describes apoptosis-induced apoptosis in imaginal discs. In that work, Eiger was identified as a death ligand. Does that apply here, too?

---

## [Decision Letter · Decision Letter 2]

13 Mar 2022

Dear Dr Karpac,

Thank you for submitting a revised version of your manuscript entitled "Synergy between Integrin-ECM interactions and membrane-associated Catalase promotes resilience of the Drosophila intestinal epithelium" for consideration as a Update Article at PLOS Biology. This revised version of your manuscript has been evaluated by the PLOS Biology editors, the Academic Editor and the three original reviewers.

As you will see, the reviewers appreciate the significant improvements you have made in the revision of the manuscript, hoewever they also raise several remaining issues that need to be addressed. To address these comments, especially those of Reviewer 2, you will need to provide a clearer description and interpretation of the "Diedel-interacting proteins" experiment in Figure 4A. You will also need to provide the controls requested by Reviewer 2 for the catalase neutralising antibody experiment in Figure S7. If this is not possible, it would be acceptable to submit a revised manuscript without the catalase neutralising antibody experiment. We also emphasize that, when revising the text, particular attention should be given to linking together properly the Diedel-integrin and extracellular catalase halves of the manuscript, including the abstract.

In light of the reviews (attached below), we are pleased to offer you the opportunity to address the remaining points from Reviewers 2 and 3 in a revised version that we anticipate should not take you very long. We will then assess your revised manuscript and your response to the reviewers' comments and we may consult the reviewers again.

We expect to receive your revised manuscript within 1 month.

**IMPORTANT - SUBMITTING YOUR REVISION**

3. Resubmission Checklist

a) *PLOS Data Policy*

b) *Published Peer Review*

Sincerely,

Ines

--

Ines Alvarez-Garcia, PhD

Senior Editor

PLOS Biology

Reviewers' comments

Rev. 1:

The authors have attempted to solve many concerns I raised during the initial round of review. Thanks to the new data added to the manuscript, now it seems more convincing to conclude what authors tried to show.

Rev. 2:

This is a re-review of the manuscript now titled "Synergy between Integrin-ECM interactions and membrane-associated Catalase promotes resilience of the Drosophila intestinal epithelium" by Mlih and Karpac.

The prior version of this manuscript was odd: the first part was a solid exploration of integrins as receptors for Diedel and the second part was a description of the claim that catalase is secreted under stress and interacts with Diedel to protect the epithelium. The connection between the two parts was tenuous. The connection between the two parts has now become more tenuous because the authors no longer think that Diedel and catalase physically interact. It is even less clear how we are to interpret the proteomic experiment in 4A given that the authors no longer believe that there is a direct physical interaction between Diedel and catalase. The figure legend here still reads "Mass spectrometry of gel-based pulldown analysis (Diedel-interacting proteins)".

One of the most surprising parts of the first manuscript was the claim that catalase is present and required in the extracellular space to protect tissue from stress. This would be very important if shown. However, the experiments in the prior version of the paper were not strong in this regard and the authors do not give a significantly stronger reason to believe that this is the case. The added functional experiment (blocking extracellular catalase with an antibody) requires controls. The antibody used would not be predicted to interact strongly with Drosophila catalase, based on the location of the epitope, and it's unclear whether it would block catalase function even if it were bound. This experiment would require that these points be clarified. It would also require control treatment with an isotype control antibody to eliminate other kinds of artifacts.

The oxidized cysteine experiments are interesting but also do not directly require extracellular catalase—this effect could be indirect.

The most obvious way for catalase to end up in the extracellular space would be fusion of peroxisomes with the plasma membrane. This would be very interesting but is not explored. The references given in support of this possibility in the response to reviewers are not salient to this question, or to the larger question of how catalase ends up outside the cell; they all deal with proteins in the ER or Golgi (known secretory compartments) being secreted in a condition or cell-type dependent way.

There may be something in the second half of this paper that is very important. It deserves to be analyzed carefully and thoroughly, not appended to a perfectly fine paper on Diedel receptors. Demonstration that integrins are Diedel receptors would appear to be a perfectly sufficient research update in its own right.

Rev. 3:

I am pleased with the revision of this manuscript. The authors have addressed all comments and weaknesses, the writing has improved and they have added new data to further support their statements. It is an interesting paper with important conclusions.

However, I have a few corrections about the statements the authors made about reference 11: In the discussion (lines567-570), the authors state that ref. 11 used genetically induced 'undead' cells. That is not true. Ref. 11 postulates that dying enterocytes in the adult midgut undergo a transiently undead state in the course of dying. This state is not genetically induced (like by coexpression of p35). Furthermore, in the same context, the authors state that ref. 11 showed that Duox (via Myo1D) localizes to the basement membrane. Again, that was not shown in ref. 11. Duox is a transmembrane protein sitting in the plasma membrane. What was shown in ref 11 is that the caspase Dronc is localized to the plasma membrane via Myo1D. These incorrect statements need to by corrected by the authors.

---

## [Editor Report · Decision Letter 3]

8 Apr 2022

Dear Dr Karpac,

Thank you for submitting your revised Update Article entitled "Synergy between Integrin-ECM interactions and membrane-associated Catalase promotes resilience of the Drosophila intestinal epithelium" for publication in PLOS Biology. I have now obtained advice from the team of editors and the Academic Editor, who has checked the revision.

Based on the reviews (attached below), we will probably accept this manuscript for publication, provided you satisfactorily address the policy-related requests stated below.

In addition, we would like you to consider a suggestion to improve the title:

"Integrin-ECM interactions and membrane-associated Catalase work together to promote resilience of the Drosophila intestinal epithelium"

We expect to receive your revised manuscript within two weeks. 

*Published Peer Review History*

*Press*

Sincerely,

Ines

--

Ines Alvarez-Garcia, PhD

Senior Editor

PLOS Biology

DATA POLICY:

thank you for providing the data underlying all the graphs shown in the figures. Please also indicate in the figure legends where the underlying data can be found - for example, you can add "The data underlying the graphs shown in the figure can be found in S1_Data"

Please add the original, uncropped blot shown in Fig. 2C

We require the original, uncropped and minimally adjusted images supporting all blot and gel results reported in an article's figures or Supporting Information files. We will require these files before a manuscript can be accepted so please prepare and upload them now. Please carefully read our guidelines for how to prepare and upload this data: https://journals.plos.org/plosbiology/s/figures#loc-blot-and-gel-reporting-requirements

---

## [Editor Report · Decision Letter 4]

19 Apr 2022

Dear Dr Karpac,

On behalf of my colleagues and the Academic Editor, Alex Gould, I am pleased to say that we can in principle accept your Update Article entitled "Integrin-ECM interactions and membrane-associated Catalase cooperate to promote resilience of the Drosophila intestinal epithelium" for publication in PLOS Biology, provided you address any remaining formatting and reporting issues. These will be detailed in an email that will follow this letter and that you will usually receive within 2-3 business days, during which time no action is required from you. Please note that we will not be able to formally accept your manuscript and schedule it for publication until you have completed any requested changes.

PRESS

Sincerely, 

Ines

--

Ines Alvarez-Garcia, PhD 

Senior Editor 

PLOS Biology
